# ATR expands embryonic stem cell fate potential in response to replication stress

Sina Atashpaz[1†]*, Sara Samadi Shams[1†], Javier Martin Gonzalez[2], Endre Sebestyén[1‡], Negar Arghavanifard[1,3], Andrea Gnocchi[1,3], Eliene Albers[4], Simone Minardi[1,5], Giovanni Faga[6], Paolo Soffientini[1], Elisa Allievi[5], Valeria Cancila[7], Angela Bachi[1], Óscar Fernández-Capetillo[8,9], Claudio Tripodo[7], Francesco Ferrari[1], Andrés Joaquin López-Contreras[4], Vincenzo Costanzo[1,3]*

[1]IFOM-The FIRC Institute of Molecular Oncology, Milan, Italy; [2]Transgenic Core Facility, University of Copenhagen, Copenhagen, Denmark; [3]Department of Oncology and Hemato-oncology, University of Milan, Milan, Italy; [4]Center for Chromosome Stability and Center for Healthy Aging, Department of Cellular and Molecular Medicine, University of Copenhagen, Copenhagen, Denmark; [5]Cogentech, IFOM-The FIRC Institute of Molecular Oncology Milan, Milan, Italy; [6]Experimental Therapeutics Program, IFOM-The FIRC Institute of Molecular Oncology, Milan, Italy; [7]Tumor Immunology Unit, Department of Health Sciences, Human Pathology Section, University of Palermo School of Medicine Palermo, Palermo, Italy; [8]Spanish National Cancer Research Center, Madrid, Spain; [9]Science for Life Laboratory, Division of Genome Biology, Department of Medical Biochemistry and Biophysics, Karolinska Institute, Stockholm, Sweden

*For correspondence:
satashpaz@gmail.com (SA);
vincenzo.costanzo@ifom.eu (VC)

†These authors contributed equally to this work

Present address: ‡1st Department of Pathology and Experimental Cancer Research, Semmelweis University, Budapest, Hungary

Competing interests: The authors declare that no competing interests exist.

**Abstract** Unrepaired DNA damage during embryonic development can be potentially inherited by a large population of cells. However, the quality control mechanisms that minimize the contribution of damaged cells to developing embryos remain poorly understood. Here, we uncovered an ATR- and CHK1-mediated transcriptional response to replication stress (RS) in mouse embryonic stem cells (ESCs) that induces genes expressed in totipotent two-cell (2C) stage embryos and 2C-like cells. This response is mediated by *Dux*, a multicopy retrogene defining the cleavage-specific transcriptional program in placental mammals. In response to RS, DUX triggers the transcription of 2C-like markers such as murine endogenous retrovirus-like elements (MERVL) and *Zscan4*. This response can also be elicited by ETAA1-mediated ATR activation in the absence of RS. ATR-mediated activation of DUX requires GRSF1-dependent post-transcriptional regulation of *Dux* mRNA. Strikingly, activation of ATR expands ESCs fate potential by extending their contribution to both embryonic and extra-embryonic tissues. These findings define a novel ATR dependent pathway involved in maintaining genome stability in developing embryos by controlling ESCs fate in response to RS.

## Introduction

ESCs are characterized by self-renewal and the ability to propagate for several cycles in vitro and in vivo (*Giachino et al., 2013*). Even if ESCs exhibit several markers of RS (*Ahuja et al., 2016*), they are able to maintain genome integrity more efficiently than differentiated cells (*Giachino et al., 2013*). The mechanisms underlying such distinctive feature are largely unknown.

ESC colonies harbor a small transient subpopulation of cells (2C-like cells) with functional and transcriptional features of totipotent 2C-stage embryos (*Choi et al., 2017*; *Ishiuchi et al., 2015*; *Macfarlan et al., 2012*). Transition to 2C-like cells has been shown to promote maintenance of

genome integrity and survival of ESCs in long-term culture (*Akiyama et al., 2015*; *Nakai-Futatsugi and Niwa, 2016*; *Zalzman et al., 2010*). In addition, several studies have demonstrated that transition to the 2C-like state confers expanded developmental potential to ESCs, making them capable of contributing to both embryonic and extra-embryonic tissues (also referred to as bidirectional cell fate potential) (*Choi et al., 2017*; *Ishiuchi et al., 2015*; *Macfarlan et al., 2012*). However, the molecular players underlying the transition to the 2C-like state in ESC culture and its possible physiological relevance in vivo in maintaining genome integrity and expanding cell fate potential in a developing embryo are not fully understood.

Here we provide several lines of evidence that ATR and CHK1-mediated response to RS triggers the activation of 2C-specific genes in ESCs and mouse embryos. This transition is hampered in ESCs derived from ATR-deficient Seckel and CHK1 haploinsufficient mouse models and following ATR or CHK1 inhibition. Significantly, we show that ETAA1-mediated ATR activation is sufficient to trigger the formation of 2C-like cells in the absence of RS.

Mechanistically, ATR-induced transition to 2C-like state is mediated by post-transcriptional regulation of the *Dux* gene, which shapes the transcriptional signature of 2C-like cells and totipotent 2C-stage embryos in placental mammals (*De Iaco et al., 2017*; *Hendrickson et al., 2017*; *Whiddon et al., 2017*). ATR-dependent regulation of *Dux* requires the GSRF1 protein, which directly binds to *Dux* mRNA promoting its stability. Importantly, activation of ATR promotes DUX-dependent formation of placental trophoblast giantlike cells (TGCs), which is hampered in ATR-deficient Seckel ESCs. Consistent with this, unlike *Dux* KO ESCs, ATR activation in WT ESCs lead to expanded cell fate potential in vivo, as shown by their ability to contribute to both inner cell mass and extra-embryonic compartments.

## Results

### RS increases the number of 2C-like cells in ESCs culture and activates the expression of 2C-like genes in mouse embryos

Maintenance of genome stability along with unlimited self-renewal is a unique feature of ESCs (*Giachino et al., 2013*). To understand how ESCs coordinate these functions, first we asked how ESCs transcriptionally respond to RS at the single cell level. To this end, we performed single cell transcriptional profiling (*Macosko et al., 2015*) of E14 mouse ESCs cultivated in Leukemia Inhibitory Factor (LIF) plus MEK and GSK inhibitors (2i) upon treatment with aphidicolin (APH), a reversible inhibitor of DNA polymerases that activates ATR by stalling replication forks progression (*Aze et al., 2016*). Unsupervised clustering analysis of CNTL and APH-treated cells (*Macosko et al., 2015*) identified a distinct subset of cells (*Figure 1a*, Cluster 4; *supplementary file 1*), that was also clearly separated by Principal Component (PC) one from the rest of the population (*Figure 1—figure supplement 1a and b*; *supplementary file 1*). The analysis of differentially expressed genes (DEGs) between cluster four and the rest of the population identified a significant enrichment of 2C-specific genes in this cluster, including *Eif1a-like* genes (*Gm5662, Gm2022, Gm4027, Gm2016*, and *Gm8300*), *Tcstv3*, and *Zscan4* genes (*Zscan4a–Zscan4d*), (*Figure 1b and c*; *Figure 1—figure supplement 1c and d*; *supplementary file 1* and *2*), the transcription of which has been shown to play a critical role in maintaining ESCs genome stability (*Akiyama et al., 2015*; *Nakai-Futatsugi and Niwa, 2016*; *Zalzman et al., 2010*). Remarkably, we found a statistically significant increase in the number of cells expressing 2C-specific markers upon RS (*Figure 1d*; *supplementary file 1*). To understand whether the increase in the number of 2C-like cells was a response to RS or it was limited to APH treatment, we exposed *pZscan4*-Emerald ESCs generated by the stable introduction of the Emerald-GFP reporter under the *Zscan4c* promoter (*Zalzman et al., 2010*) to a range of RS-inducing agents, including APH, hydroxyurea (HU) and ultraviolet light (UV) (*Cimprich and Cortez, 2008*). Fluorescence Activated Cell Sorting (FACS) analysis confirmed a significant increase in the number of Emerald positive (Em$^+$) ESCs across all treated conditions in a dose-dependent manner (*Figure 1e*; *Figure 1—source data 1*). Of note, the increase in the number of Em$^+$ ESCs upon short exposure to UV revealed that the continuous presence of the RS-inducing agent was not necessary for the activation of 2C-like cells (*Figure 1e*; *Figure 1—source data 1*), suggesting that replication fork stalling induced by UV-mediated DNA lesions was sufficient to trigger this pathway. Next, to uncover the minimum timing required for the activation of ZSCAN4, we performed a time-course

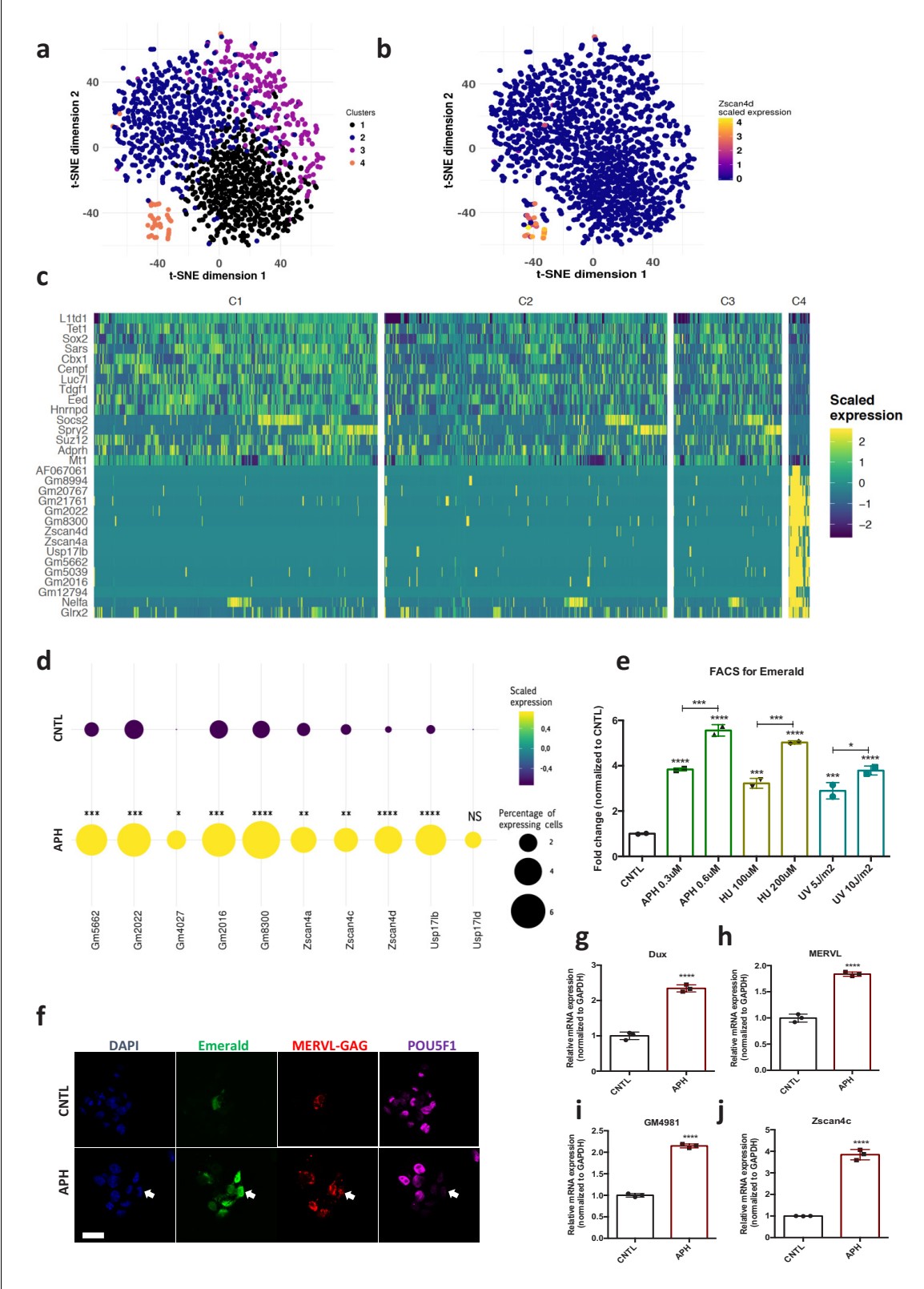

**Figure 1.** Induction of RS increases the number of 2C-like cells in ESCs culture and activates the expression of 2C-specific genes in mouse embryos. (a) Clustering of 1399 Drop-seq single-cell expression profiles into four cell populations. The plot shows a two-dimensional representation (t-SNE) of global gene expression relationship; cells are colored according to their cluster. Clusters were identified by shared nearest neighbor algorithm (see methods). (b) t-SNE plot showing *Zscan4d* expression level across all CNTL and APH-treated cells. (c) Heatmap showing the list of top 30 genes that are

*Figure 1 continued on next page*

*Figure 1 continued*

differentially expressed between cluster 4 cells (orange cluster in a) and the rest of the population (cluster 1,2, and 3). (d) Plot showing the scaled expression of 2C-specific markers and the percentage of cells expressing 2C-related genes in CNTL and APH-treated condition. Fisher's exact test was used to determine p-values. (e) FACS analysis on p*Zscan4*-Emerald ESCs upon treatment with various RS-inducing agents. (f) Immunostaining of ESCs for ZSCAN4-Emerald, MERVL-GAG and canonical pluripotency marker POU5F1 upon treatment with APH (bar = 25 μm). (g–j) RT-qPCR analysis on blastocyst-stage embryos treated with APH for key 2C-like markers. Statistical significance compared to CNTL unless otherwise indicated. All bar-plots show mean with ± SD (*p≤0.05, **p≤0.01, ***p≤0.001, ****p≤0.0001, one-way ANOVA).

The online version of this article includes the following source data and figure supplement(s) for figure 1:

**Source data 1.** FACS, qPCR and Western quantification.
**Figure supplement 1.** Induction of RS increases the number of 2C-like cells in ESCs culture.
**Figure supplement 2.** Characterization of APH-induced 2C-like cells.
**Figure supplement 3.** Characterization of APH-induced 2C-like cells.

experiment in which ESCs were treated with APH and subsequently harvested at various time points. Immunoblot on ESCs revealed a rapid phosphorylation of CHK1 while a significant increase in ZSCAN4 protein took place after 4–8 hr of treatment with APH (*Figure 1—figure supplement 1e*; *Figure 1—source data 1*). Importantly, although APH, mostly at a high concentration, induced mild cell apoptosis, the majority of Em$^+$ cells were not co-stained with the apoptosis marker CASPASE-3, suggesting that the emergence of Em$^+$ cells upon APH treatment was not due to the activation of apoptosis (*Figure 1—figure supplement 1f*). Moreover, no sign of cellular senescence was detected by β-galactosidase staining upon APH treatment in ESCs (*Figure 1—figure supplement 1g*).

Due to the key role of DUX and MERVL-derived long terminal repeat (LTR) elements in shaping the transcriptional signature of 2C-stage embryos and 2C-like cells (*Choi et al., 2017*; *Ishiuchi et al., 2015*; *Macfarlan et al., 2012*; *De Iaco et al., 2017*; *Hendrickson et al., 2017*; *Whiddon et al., 2017*), we next asked whether RS-induced expression of the *Zscan4* gene was accompanied by the activation of these retroelements. The RT-qPCR results revealed a linear correlation between concentration of RS inducing agents and the expression of key 2C-like markers, such as *Dux*, MERVL, *Zscan4d*, *Tcstv3*, *Gm12794*, *Gm4340* (*Figure 1—figure supplement 2a–f*; *Figure 1—source data 1*). These results were further validated using *p*MERVL-GFP ESCs (ESCs expressing a GFP reporter under the control of MERVL promoter *Ishiuchi et al., 2015*; *De Iaco et al., 2017*; *Figure 1—figure supplement 2g and h*; *Figure 1—source data 1*). However, induction of RS did not affect the expression of main 2C-specific markers such as MERVL and *Zscan4* (no expression was detected by qPCR) in two mouse embryonic fibroblast (MEF) lines with distinct genetic backgrounds (C57BL/6J and 129P2/OlaHsd) (*Figure 1—figure supplement 2i*; *Figure 1—source data 1*), suggesting the involvement of alternative repressive mechanisms that suppress the activation of 2C-like transcriptional program in more differentiated cells.

2C-like cells reduce the expression of pluripotency markers at protein but not transcriptional level (*Choi et al., 2017*; *Ishiuchi et al., 2015*; *Macfarlan et al., 2012*). Thus, to understand whether RS-induced cells share such feature with 2C-like cells, we monitored the expression of canonical pluripotency markers upon APH treatment. In agreement with previous findings (*Choi et al., 2017*; *Ishiuchi et al., 2015*), we detected downregulation of SOX2, and POU5F1 (also known as OCT4) proteins in ZSCAN4$^+$/Em$^+$ cells by immunostaining and immunoblotting (*Figure 1f*; *Figure 1—figure supplement 2j and k*; *Figure 1—source data 1*). However, no significant alteration in the expression of pluripotency-related genes (e.g., *Nanog*, *Oct4* and *Rex1*) was observed at the transcriptional level (*Figure 1—figure supplement 2l–n*; *Figure 1—source data 1*). As previously reported (*Ishiuchi et al., 2015*), 2C- like cells that were identified by expression of 2C markers, MERVL-GAG and ZSCAN4, and the absence of OCT4 protein, were found to lack chromocenters (*Figure 1—figure supplement 2j*).

The ZSCAN4$^+$/MERVL$^+$ cells were reported to be present in all phases of the cell cycle albeit with higher percentage in G2/M phase (*Eckersley-Maslin et al., 2016*). Of note, the cell cycle analysis on p*Zscan4*-Emerald ESCs confirmed that APH treatment neither at low (0.3 μM) nor at high (6 μM) concentration could increase the G2/M population in culture (*Figure 1—figure supplement 3a*). Similar results were obtained upon exposure of ESCs to UV (*Figure 1—figure supplement 3b*), indicating that the increase in the Em$^+$ population upon APH and UV treatment was not due to cell cycle arrest in G2/M. This is consistent with a previous work showing that G2/M arrest by nocodazole is not

sufficient to trigger *Zscan4* expression (*Storm et al., 2014*). Finally, to evaluate the physiological relevance of these findings in vivo, we asked whether transient induction of RS could activate the expression of key 2C-embryo specific genes at the later stages of mouse embryonic development. To this end, morula-stage embryos were treated with APH and subsequently, the synchronized embryos were selected and subjected to RT-qPCR. Strikingly, induction of RS activated the expression of several key 2C-embryo specific markers, including *Zscan4c*, *Dux*, MERVL and *Gm4981* in mouse embryos (*Figure 1g–j*; *Figure 1—figure supplement 3c–h*; *Figure 1—source data 1*). Importantly, APH treatment did not disrupt the blastocyst formation or expression of ICM and TE markers (*Figure 1—figure supplement 3i–k*).

Overall, these findings indicate that RS leads to the activation of 2C-embryo specific genes in ESCs and morula-stage mouse embryos.

## ATR and CHK1-mediated RS response triggers the activation of key 2C-like genes in ESCs

Next, to gain further insight into the mechanisms through which RS response (RSR) could contribute to the emergence of 2C-like cells in ESCs culture, we asked whether activation of the endogenous DNA damage response pathways (DDR) is responsible for the emergence of 2C-like cells under normal culture condition. To this end, we inhibited DDR pathways by treating p*Zscan4*-Emerald ESCs with specific ATM and ATR inhibitors (ATMi, KU-55933 and ATRi, VE 822, respectively). FACS analysis revealed a slight reduction in the number of Em$^+$ cells upon ATR inhibition; however, treatment with ATMi did not significantly reduce the fraction of Em$^+$ cells in culture (*Figure 2a*; *Figure 2—figure supplement 1a*; *Figure 2—source data 1*). In addition, FACS analysis revealed a significant enrichment of γH2AX$^+$ and p-CHK1$^+$ (indicators of RS) populations within Em$^+$ ESCs (*Figure 2—figure supplement 1 c and d*; *Figure 2—source data 1*), suggesting that the transient activation of ZSCAN4 is linked to the presence of endogenous RS in 2C-like cells. Consistent with these results, inhibition of ATR but not ATM activity could robustly revert the expression of ZSCAN4 and *Dux*, which was induced in response to various RS-inducing agents (e.g., UV, HU and APH) (*Figure 2a and b*; *Figure 2—figure supplement 1 a, b, e and f*; *Figure 2—source data 1*). These results were further validated on E14 and R1 ESC lines through RT-qPCR assay for *Zscan4d* and MERVL (*Figure 2c*; *Figure 2—figure supplement 1g*; *Figure 2—source data 1*). Similar results were obtained in ATM KO ESCs confirming that activation of APH-induced 2C-like genes in ESC culture is not ATM dependent (*Figure 2—figure supplement 1h–j*; *Figure 2—source data 1*). Of note, inhibition of p38 MAPK signaling pathway did not lead to any decrease in the level of APH-induced 2C-gene expression. This result suggests that unlike the role of p38 inhibitors in suppressing DUX4 in cellular and animal models of facioscapulohumeral muscular dystrophy (*Oliva et al., 2019*), this pathway does not play a major role in the regulation of *Dux* upon RS induction in mouse ESC (*Figure 2—figure supplement 1k*; *Figure 2—source data 1*).

In agreement with our findings, single cell transcriptional profiling revealed that suppression of ATR activity by ATRi led to a statistically significant reduction in the number of cells that express main 2C-genes in response to APH (e.g., *Zscan4d$^+$*, *Gm8300$^+$*, *Gm022$^+$*, and *Usp14ib$^+$*) (*Figure 2d*). However, we did not find any significant change in the expression level of 2C-specific genes at the single cell level across different conditions (i.e., CNTL, APH and APH+ATRi) (*Figure 2—figure supplement 1l–n*). Moreover, FACS analysis of p*MERVL*-GFP ESCs did not show any significant increase in the mean intensity of MERVL-GFP signal within MERVL$^+$ population between CNTL and APH-treated conditions (*Figure 1—figure supplement 2g and h*; *Figure 1—source data 1*). These results suggest that the increase in the expression of 2C-genes upon APH treatment is mainly due to the increase in the number of newly generated 2C-like cells and not to the overexpression of 2C-genes in the pre-existing 2C-like cell population.

To further validate these findings, we derived ATR-deficient Seckel (*Atr$^{Sec/Sec}$*) and haploinsufficient (*Chk1$^{+/-}$*) ESCs from previously reported mouse models (*Liu et al., 2000*; *Murga et al., 2009*) as the complete ablation of ATR or CHK1 causes embryonic lethality in mice (*Brown and Baltimore, 2000*). To this end, ESC lines were established in culture using pre-implantation embryos obtained from crosses between heterozygous mice either for *Atr* Seckel or for *Chk1* KO alleles. On the basis of genotyping results, homozygous *Atr* Seckel and heterozygous *Chk1* ESC lines were characterized for further investigations (*Figure 2e and f*; *Figure 2—figure supplement 2a and b*; *Figure 2—source data 1*). Although immunoblot results confirmed that the levels of ATR and CHK1 proteins

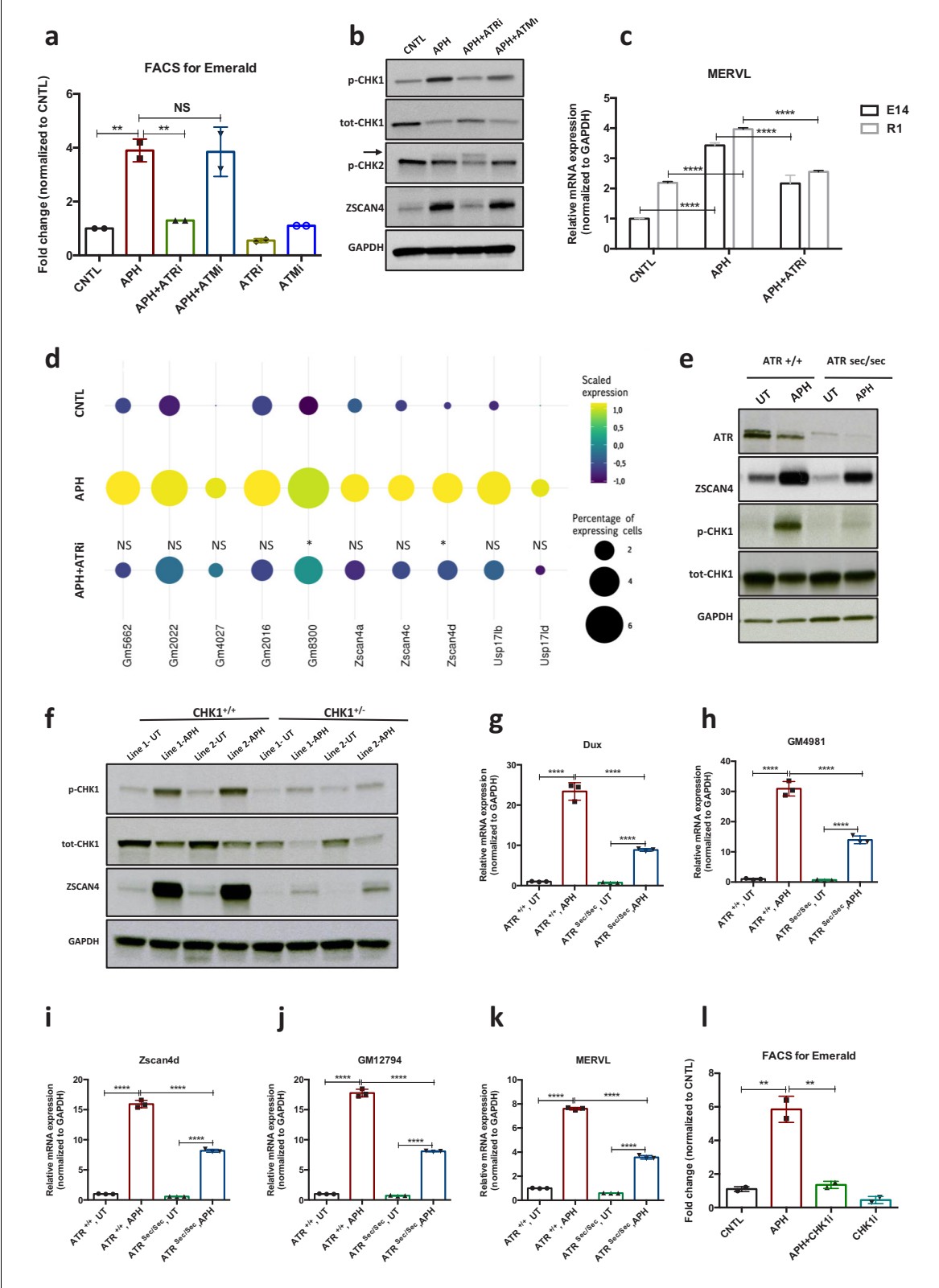

**Figure 2.** ATR and CHK1-mediated RSR triggers activation of key 2C-specific genes in ESCs. (a) FACS analysis on p*Zscan4*-Emerald ESCs showing the number of Em+ cells upon treatment with APH and specific ATR and ATM inhibitors. (b) Immunoblot for the phosphorylation status of key DDR kinases (CHK1 and CHK2) and the ZSCAN4 protein level upon treatment with APH and ATM/ATR inhibitor in ESCs. (c) RT-qPCR analysis of two ESCs lines for the expression of MERVL upon treatment with APH and ATRi. (d) Plot showing the scaled expression of 2C-specific markers and the percentage of cells

*Figure 2 continued on next page*

*Figure 2 continued*

expressing 2C-related genes in CNTL, APH-treated and APH+ATRi conditions. Fisher's exact test was used to determine p-values. (**e**) Immunoblot for ZSCAN4, ATR and the phosphorylation status of CHK1 upon APH treatment of $Atr^{Sec/Sec}$ and $Atr^{+/+}$ ESCs. (**f**) Immunoblot showing the expression of ZSCAN4 and p-CHK1 in $Chk1^{+/-}$ and $Chk1^{+/+}$ ESCs upon treatment with APH. (**g–k**) RT-qPCR for 2C-specific genes in $Atr^{Sec/Sec}$ and $Atr^{+/+}$ ESCs treated with APH. (**l**) FACS analysis of p*Zscan4*-Emerald ESCs showing the number of Em$^+$ cells upon treatment with APH and a specific CHK1 inhibitor. Statistical significance compared to CNTL unless otherwise indicated. All bar plots show mean with ± SD (*p≤0.05, **p≤0.01, ***p≤0.001, ****p≤0.0001, one-way ANOVA). For western blots quantification refer to *Figure 2—source data 1*.

The online version of this article includes the following source data and figure supplement(s) for figure 2:

**Source data 1.** FACS, qPCR and Western quantification.
**Figure supplement 1.** ATR-mediated RSR triggers activation of key 2C-specific genes in ESCs.
**Figure supplement 2.** ATR and CHK1-mediated RSR triggers activation of key 2C-specific genes in ESCs.

were severely reduced in comparison to the wild type (WT) ESCs (*Figure 2e and f*; *Figure 2—source data 1*), no significant alteration in the expression of key pluripotency genes was observed in $Atr^{Sec/Sec}$ or $Chk1^{+/-}$ ESCs compared to WT ESCs (*Figure 2—figure supplement 2a and b*). Noticeably, similar to the inhibitory impact of ATRi on the expression of 2C-genes, APH treatment on $Atr^{Sec/Sec}$ ESCs led to only a mild increase in the expression of 2C-related genes unlike WT ESCs (*Figure 2e and g–k*; *Figure 2—figure supplement 2c–f*; *Figure 2—source data 1*). As expected, similar results were obtained with $Chk1^{+/-}$ ESCs and upon treatment with CHK1 inhibitor (CHK1i) (*Figure 2f and l*; *Figure 2—figure supplement 2g–n*; *Figure 2—source data 1*). Of note, we did not find any alteration in the level of APH-induced *Dux* upon *Trp53* knockdown (KD) or its complete ablation in *Trp53* KO ESCs, suggesting that activation of 2C-like pathway does not require the known mediator of DDR, P53 (*Figure 2—figure supplement 2o–r*; *Figure 2—source data 1*).

Overall, these data indicate that the ATR and CHK1-stimulated response to RS regulates the activation of 2C-specific genes.

## ATR induces the transcriptional signature of 2C-like cells in ESCs

Next, to understand whether activation of ATR-dependent response could result in a global transcriptional activation of 2C-specific genes, we performed high-throughput transcriptional profiling on three ESC lines, namely E14, R1 and MC1, upon APH treatment. Analysis of differentially expressed genes (DEGs) (FDR < 0.05 and |log2 fold change| > 1) identified 3074 upregulated genes with more than two-fold change in gene expression upon APH treatment, and only 640 downregulated genes (*Figure 3—figure supplement 1a*; *supplementary file 3*), which is in agreement with the general openness of chromatin in 2C-like cells (*Ishiuchi et al., 2015*). To understand how many of the identified DEGs overlap with those specifically expressed in 2C-like cells, we compared our list of APH-induced genes with a recently published dataset (*Eckersley-Maslin et al., 2016*). Through such comparison, we found that a significant fraction of APH-induced retroviral elements and genes overlap with those expressed in 2C-like cells, including MERVL, MT2_Mm, *Dux*, *Eif1a-like* genes (*Gm5662*, *Gm2022*, *Gm4027*, and *Gm8300*), Zscan4 genes (*Zscan4b* and *Zscan4d*), *Zfp352*, *Zfp750*, *Tdpoz* genes (*Tdpoz1* and *Tdpoz3*), and *Tmem92* (*Figure 3a and b*; *Figure 3—figure supplement 1b*; *supplementary file 4* and *5*). Considerably, RT-qPCR results confirmed the significant upregulation of main retroviral elements and 2C-specific genes upon APH treatment (*Figure 3—figure supplement 1c–k*; *Figure 3—source data 1*). Interestingly, the DEG analysis identified a large portion (48%) of APH-induced 2C-specific genes to be transcriptionally repressed upon ATR inhibition (*Figure 3c*; *supplementary file 4*).

Recent reports demonstrated that 2C-like cells can be generated through genetic modulation of several factors, including the KD of chromatin assembly factor-1 (CAF-1), KD of KRAB (Kruppel-Associated Box Domain)-Associated Protein 1 (KAP1), KO of microRNA-34 (miR-34) and OE of DUX (*Choi et al., 2017*; *Ishiuchi et al., 2015*; *Hendrickson et al., 2017*; *Rowe et al., 2010*). Thus, to test whether these factors were involved in ATR-induced expression of 2C-related genes, we compared the transcriptome of APH-induced ESCs with those from published datasets. While we found a significant overlap with all datasets, the highest number of overlapping genes was found with the transcriptome of CAF-1 KD ESCs (*Figure 3—figure supplement 1l*; *supplementary file 4*), possibly due to the previously reported role of CAF-1 in preventing RS (*Hoek and Stillman, 2003*).

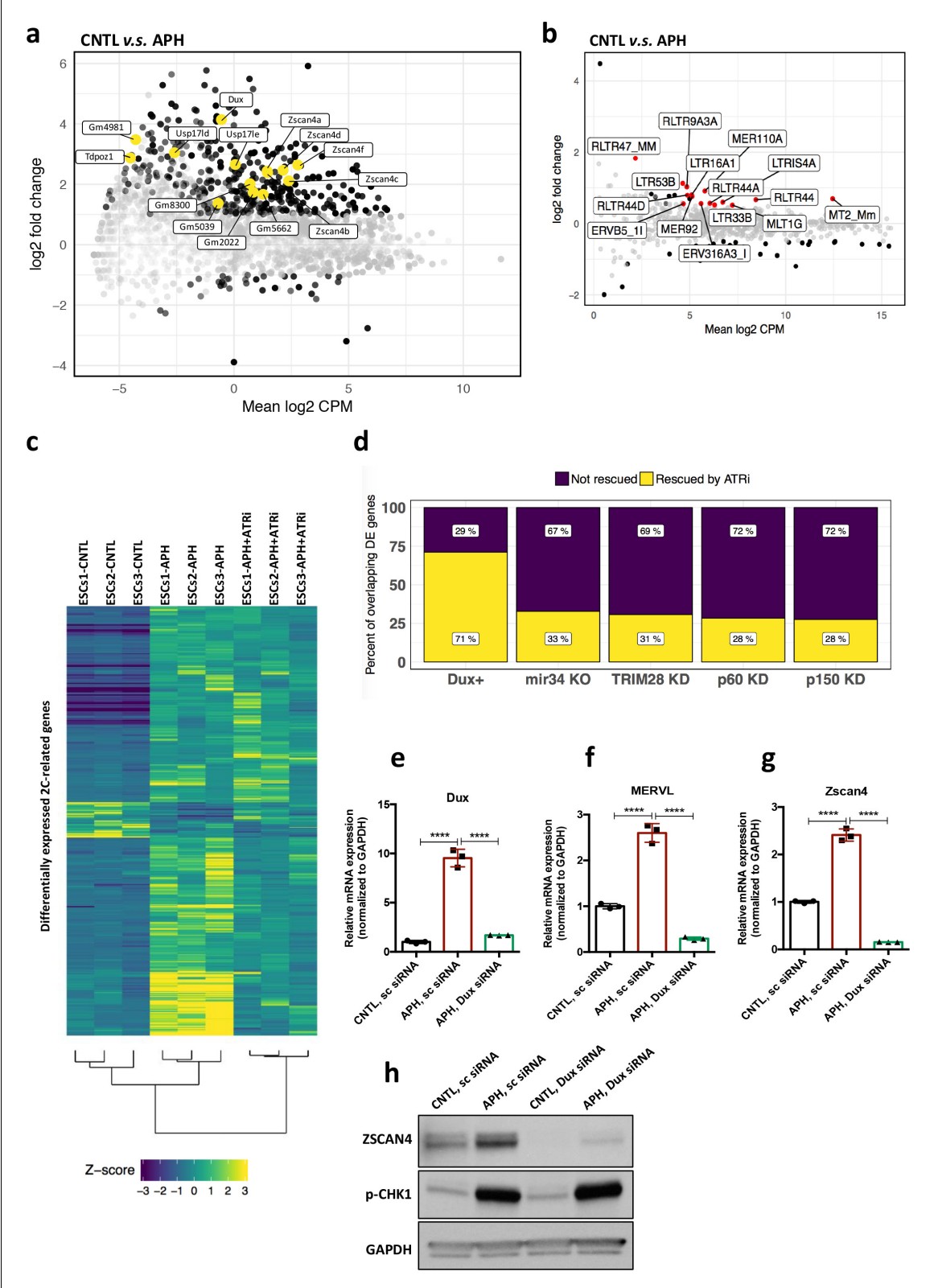

**Figure 3.** ATR induces transcriptional signature of 2C-like cells in ESCs. (a) MA plot showing gene expression in ESCs treated with APH in comparison to control. Key 2C-specific genes are highlighted (b) MA plot showing retrotransposons expression upon APH treatment. (c) Heatmap showing the robust z-scores for 2C-specific genes in the indicated samples. 2C-related genes were identified by performing a differential expression analysis on ZSCAN4⁺/MERVL⁺ v.s. ZSCAN4⁻/ MERVL⁻ ESCs from *Eckersley-Maslin et al. (2016)*. (d) Bar plot displaying the percentage of ATR-dependent

*Figure 3 continued on next page*

*Figure 3 continued*

differentially expressed genes among the ones shared between APH-treated ESCs and each dataset. (**e–g**) RT-qPCR analysis of *Dux* KD ESCs for *Dux* and *Zscan4d* genes, and MERVL upon treatment with APH. (**h**) Immunoblot for p-CHK1 and ZSCAN4 proteins upon treatment with APH in Dux KD ESCs in comparison with control ESCs. Statistical significance compared to CNTL unless otherwise indicated. All bar plots show mean with ± SD (*$p \leq 0.05$, **$p \leq 0.01$, ***$p \leq 0.001$, ****$p \leq 0.0001$, one-way ANOVA). For western blots quantification refer to *Figure 3—source data 1*.
The online version of this article includes the following source data and figure supplement(s) for figure 3:

**Source data 1.** qPCR and Western quantification.
**Figure supplement 1.** ATR induces transcriptional signature of 2C-like cells in ESCs.
**Figure supplement 2.** ATR induces transcriptional signature of 2C-like cells in ESCs.

Next, to uncover the specific role of ATR in controlling 2C-genes regulating factors, we focused our analysis on genes whose expression was reverted by ATRi. Remarkably, through such analysis we found that 71% of the genes shared between APH and DUX-induced conditions were rescued by ATRi (*Figure 3d*, *supplementary file 4*), suggesting a possible role of DUX in activation of 2C-specific genes through ATR. To validate this finding, we checked the expression of key 2C-related genes in *Dux* KO ESCs after induction of RS. Importantly, although APH treatment, through ATR activation, increased the expression of *Dux* and its downstream targets, such as *Zscan4* and MERVL in WT ESCs, it could not induce the expression of key 2C-genes in *Dux* KO ESCs (*Figure 3—figure supplement 2a–c*; *Figure 3—source data 1*). Next, to understand whether the inability of APH in activating MERVL and *Zscan4* is due to a general suppression of these genes in *Dux* KO ESCs or not, we performed a siRNA-mediated KD experiment to partially silence *Dux* in WT ESCs. As expected, upon partial *Dux* KD in CNTL condition, the expression level of *Zscan4* and MERVL decreased but was not fully abolished (*Figure 3—figure supplement 2d–f*; *Figure 3—source data 1*). Consistent with our previous findings, RT-qPCR results confirmed that the activation of ATR through APH treatment increases the expression of 2C-genes in WT ESCs, while APH could only modestly activate *Zscan4* and MERVL in *Dux* KD ESCs (*Figure 3e–g*; *Figure 3—source data 1*). These results were further validated by immunoblot for ZSCAN4 (*Figure 3h*; *Figure 3—source data 1*) and confirmed the role of ATR in regulating 2C-genes through *Dux* activation.

Overall, these findings suggest that ATR is a potent upstream driver of 2C-genes expression in ESCs.

## ETAA1-mediated activation of ATR elevates the number of 2C-like cells in a RS-free context

Recent evidence showed that ATR kinase can be directly activated by the RPA-binding protein, Ewing's tumor-associated antigen 1 (ETAA1) in the absence of RS (*Bass et al., 2016*; *Haahr et al., 2016*). Hence, to confirm that the activation of 2C-specific genes is due to ATR-mediated response and not the physical damage to DNA, we aimed to activate ATR in a DNA damage-free context through overexpression of ETAA1-ATR activating domain (ETAA1-AAD) in ESCs. To this end, p*Zscan4*-Emerald ESCs were infected with two independent lentiviruses generated from two different clones of ETAA1-expressing lentivectors in which ETAA1-AAD was expressed under the control of a doxycycline (Dox)-inducible promoter, and subsequently the cells were selected against puromycin and neomycin for two weeks. As expected, FACS analysis confirmed the activation of the ATR pathway, as shown by an increase in the number of γH2AX$^+$ cells upon Dox administration to ETAA1-AAD-inducible ESCs (Dox-iETAA1 ESCs) compared to ESCs infected with an empty vector (EV) (*Figure 4—figure supplement 1a*). However, ETAA1-AAD overexpression did not induce cell apoptosis as shown by CASPASE-3 FACS analysis (*Figure 4—figure supplement 1b*). Strikingly, ETAA1-AAD-stimulated activation of ATR was accompanied by a significant increase of the Em$^+$ population in ESCs culture and transcriptional activation of *Zscan4, Dux* and MERVL (*Figure 4a–d*; *Figure 4—figure supplement 1* a and c-f; *Figure 4—source data 1*). As expected, this transcriptional activation was fully abolished upon ATR and CHK1 inhibition (*Figure 4a–c*; *Figure 4—figure supplement 1f–j*; *Figure 4—source data 1*).

Importantly, several lines of evidence demonstrated that the γH2AX positivity caused by the overexpression of ETAA1 is not a consequence of DNA breakage (*Bass et al., 2016*; *Haahr et al., 2016*). Consistent with this, ATR activation through ETAA1-AAD expression led to the mild phosphorylation

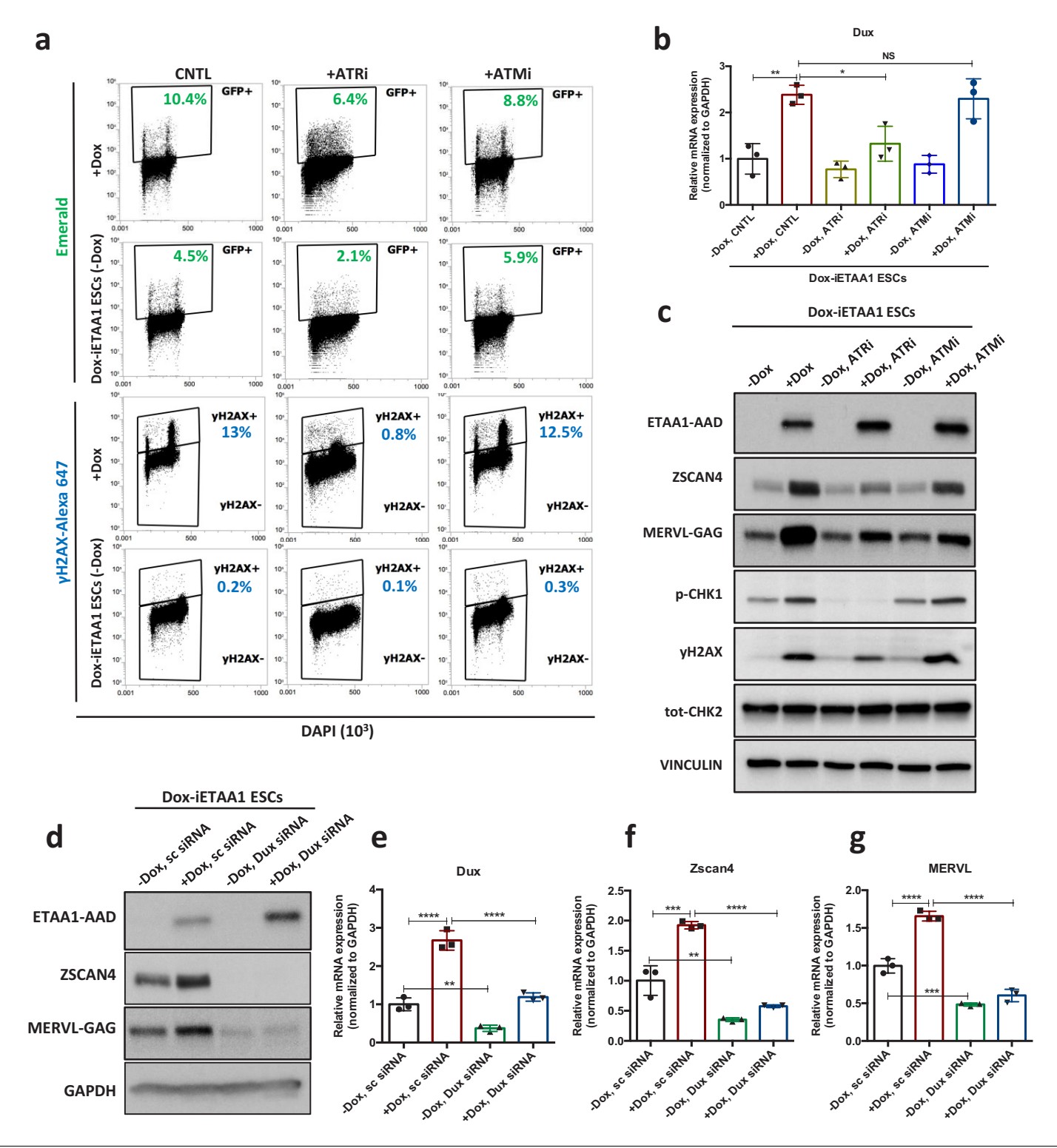

**Figure 4.** ETAA1-mediated activation of ATR induces 2C-like cells in a RS-free context. (**a**) FACS analysis for γH2AX and Emerald-GFP in Dox-iETAA1 ESCs in the presence or absence of Dox and upon treatment with ATRi or ATMi. (**b**) RT-qPCR results for *Dux* expression in Dox-iETAA1 ESCs upon Dox induction in the presence or absence of ATRi or ATMi. (**c**) Immunoblot showing the expression of ETAA1-AAD, ZSCAN4, MERVL-GAG and the phosphorylation status of CHK1, CHK2 and H2AX in Dox-iETAA1 ESCs upon treatment with Dox, ATRi or ATMi. (**d**) Immunoblot showing the expression of ETAA1-AAD, ZSCAN4, MERVL-GAG in Dox-iETAA1 ESCs upon treatment with Dox and *Dux* knock down. (**e–g**) RT-qPCR analysis of Dox-iETAA1

*Figure 4 continued on next page*

*Figure 4 continued*

ESCs for expression of 2C-related genes (*Dux*, MERVL and *Zscan4*) upon treatment with Dox and *Dux* knock down. Statistical significance compared to CNTL unless otherwise indicated. All bar plots show mean with ± SD (*p≤0.05, **p≤0.01, ***p≤0.001, ****p≤0.0001, one-way ANOVA). For western blots quantification refer to *Figure 4—source data 1*.

The online version of this article includes the following source data and figure supplement(s) for figure 4:

**Source data 1.** qPCR and Western quantification.
**Figure supplement 1.** Characterization of ETAA1-AAD inducible ESCs.
**Figure supplement 2.** ETAA1-mediated activation of ATR induces 2C-like cells in a RS-free context.

---

of H2AX and CHK1 but not CHK2 in ESCs (*Figure 4a and c*; *Figure 4—source data 1*). Moreover, while inhibition of ATR activity reduced the level of γH2AX upon ETAA1-AAD-mediated activation of ATR, inhibition of ATM activity did not exert any noticeable effect (*Figure 4a and c*; *Figure 4—source data 1*). Finally, we demonstrated that the activation of canonical 2C-genes upon ETAA1-AAD-mediated ATR activation is also DUX dependent, as *Dux* KD significantly reduced the ETAA1-AAD-mediated expression of MERVL and ZSCAN4 (*Figure 4d–g*; *Figure 4—figure supplement 2e–h*; *Figure 4—source data 1*). Similar results were obtained upon ETAA1-AAD-mediated ATR activation in *Dux* KO ESCs (*Figure 4—figure supplement 1k–m*; *Figure 4—source data 1*).

Finally, to further dissect the physiological role of ETAA1 in the transition to 2C-like state, we asked whether its KD through specific siRNA (*Figure 4—figure supplement 2a*; *Figure 4—source data 1*) could lead to a reduction in the level of 2C-specific genes expressed under basal condition. Of note, the KD of ETAA1 did not lead to a reduction of 2C-genes expression (*Figure 4—figure supplement 2a-c*; *Figure 4—source data 1*) in line with recent works where ETAA1 was found to function in parallel to TOPBP1 in regulating ATR and maintaining genome stability (*Bass et al., 2016*; *Haahr et al., 2016*). Moreover, we found a higher level of ETAA1 in 2C-like cells further confirming that activation of RSR pathways could lead to the expression of 2C genes (*Figure 4—figure supplement 2d*).

These results collectively indicate that the DNA damage-independent, ETAA1-stimulated ATR activation in ESCs is sufficient to activate 2C-related genes through DUX.

## Candidate-based screening identifies the involvement of the GRSF1 in the post-transcriptional regulation of the *Dux* mRNA downstream of ATR

Next, to identify the mediating factors regulating *Dux* expression in response to ATR activation, we performed a siRNA-based RT-qPCR screening using a library targeting 148 genes (*Figure 5a*) and quantified the expression level of *Zscan4*, a *bona fide* DUX downstream target, as the final readout (*De Iaco et al., 2017*; *Hendrickson et al., 2017*; *Whiddon et al., 2017*). To this aim, we took advantage of the MISSION esiRNA technology that provides a heterogeneous mixture of siRNAs targeting the same mRNA sequence, and thus offers a highly specific and effective gene-silencing approach at lower concentration, ensuring a minimal risk of off-target effects (*Theis and Buchholz, 2010*). We refined the list of siRNAs based on the overlap of the following categories: potential substrate of ATR/CHK1 (*Matsuoka et al., 2007*; *Blasius et al., 2011*), *Dux* promoter-bound factors based on our in silico analysis and previous findings (*Campbell et al., 2018*), and potential activators or suppressors of *Zscan4* (*Figure 5b*; *Rodriguez-Terrones et al., 2018*). We included siRNAs targeting *Dux*, *Atr*, and *Zscan4* as positive controls and a siRNA targeting Renilla luciferase as a negative control.

To identify *Dux* activators linked to ATR, we focused on genes whose KD could decrease the APH-induced *Zscan4* expression, while genes whose KD was sufficient to induce *Zscan4* expression were considered as *Dux* suppressors. Through analyzing RT-qPCR results, we identified 49 hits whose downregulation altered *Zscan4* expression at least two-fold. Interestingly, these genes were found to be enriched in two main categories: i) genes involved in DNA replication and RSR such as *Vcp*, *Smc1a*, *Rfwd3*, *Rfc2* that further validated our finding on the involvement of RSR pathway in the regulation of *Dux*; and ii) mRNA processing factors (*Figure 5c*; *supplementary file 6*), suggesting the possible regulation of *Dux* at post-transcriptional level.

The significant enrichment of mRNA processing factors in our screening analysis, led us to test whether ATR could directly affect *Dux* mRNA level. To this end, we first checked whether ATRi could

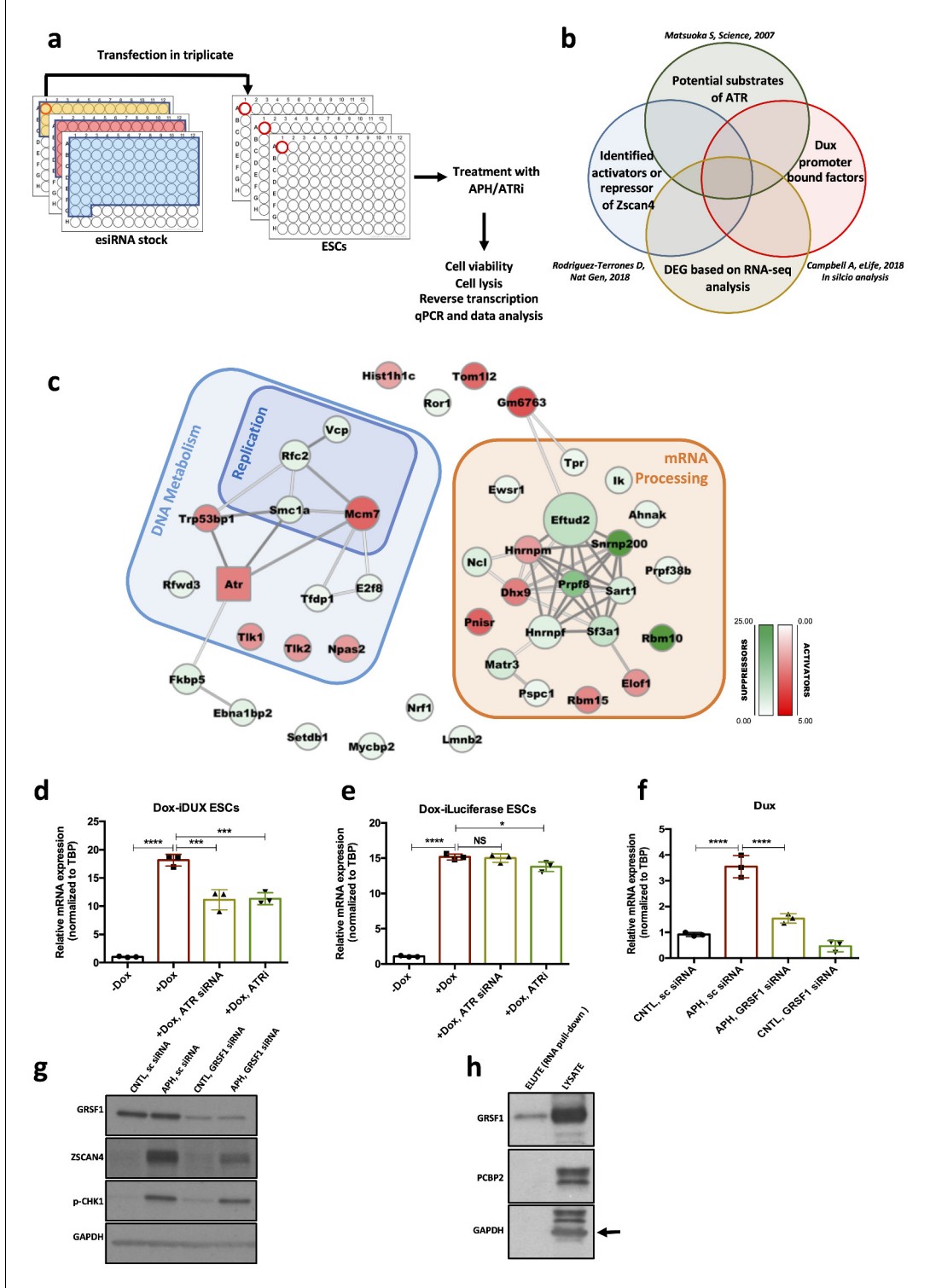

**Figure 5.** Identification of RSR downstream molecular players regulating the 2C-like state. (**a.b**) Schematic design of the esiRNA-based knock-down screening and esiRNA library selection. (**c**) Protein interaction network for the hits identified through the esiRNA screening. Activators and suppressors are highlighted in red and green, respectively. The interactions are based on the STRING database. (**d**) RT-qPCR analysis of iDox-*Dux* ESCs for exogenous *Dux* mRNA upon treatment with Dox. (**e**) RT-qPCR analysis of iDox-Luciferase ESCs for Luciferase mRNA upon treatment with Dox. (**f**) RT-qPCR analysis of *Dux* mRNA upon APH treatment and *Grsf1* KD. (**g**) Immunoblot showing the expression of GRSF1, ZSCAN4 and pCHK1 upon *Gsrf1* KD. (**h**) Immunoblot showing the

*Figure 5 continued on next page*

*Figure 5 continued*

binding of GRSF1 protein to the *Dux* mRNA. Statistical significance compared to CNTL unless otherwise indicated. All bar plots show mean with ± SD (*p≤0.05, **p≤0.01, ***p≤0.001, ****p≤0.0001, one-way ANOVA). For western blots quantification refer to *Figure 5—source data 1*.

The online version of this article includes the following source data and figure supplement(s) for figure 5:

**Source data 1.** qPCR and Western quantification.
**Figure supplement 1.** mRNA processing factors regulate the level of Dux transcript.

---

alter the level of *Dux* mRNA produced exogenously by a vector that did not contain *Dux* natural promoter. To verify this, we treated *Dux* overexpressing ESCs, generated by the stable integration of a Dox-inducible *Dux* construct into ESCs (Dox-i*Dux* ESCs), with ATRi and probed the level of exogenous *Dux* transcript using oligos specifically recognizing this construct. While Dox administration led to a remarkable increase in the level of exogenous *Dux* mRNA, ATR inhibition significantly reduced the Dox-induced *Dux* transcript (*Figure 5d*; *Figure 5—source data 1*). Similar results were obtained upon *Atr* KD, however, neither ATRi nor *Atr* KD could dramatically alter the levels of exogenous Luciferase mRNA induced by Dox administration (*Figure 5e*; *Figure 5—source data 1*). Finally, to exclude any possible effect of ATR inhibition on endogenous *Dux* gene expression, we generated a Dox-inducible *Dux* OE system using *Dux* KO ESCs in which the endogenous *Dux* genomic region was completely deleted (*De Iaco et al., 2017*), and obtained a similar result on the expression of endogenous *Dux* mRNA upon ATR inhibition (*Figure 5—figure supplement 1a*; *Figure 5—source data 1*). Importantly, *Dux* overexpression did not lead to a significant increase in the level of p-CHK1, ruling out the possibility of DUX activating RSR pathway through the ATR-CHK1 axis (*Figure 5—figure supplement 1b*; *Figure 5—source data 1*). Overall these results suggest that ATR affects the level of *Dux* mRNA through its post-transcriptional regulation.

Next, to identify the potential *Dux* mRNA-binding factor(s) through which ATR could alter *Dux* mRNA level, we performed RNA-protein pull-down using a synthetic 3' untranslated region of *Dux* mRNA (*Dux* 3'UTR). Mass-spectrometry analysis of the pulled-down proteins revealed several *Dux* 3'UTR binding factors, including RNA binding proteins HNRNPA1, HNRNPA3, HNRNPA2B1, PABPN1, PABPC1, PCBP1, PCBP2 and GRSF1 (*supplementary file 7*). Strikingly, although both *Grsf1* and *Pcbp2* KD could reduce APH-induced *Dux* expression (*Figure 5f and g*; *Figure 5—figure supplement 1c*; *Figure 5—source data 1*), only GRSF1, which belongs to a group of heterogeneous nuclear RNPs bearing the RNA-recognizing domain RRM (*Ufer, 2012*), was proven to bind *Dux* 3'UTR by immunoblot (*Figure 5h*; *Figure 5—source data 1*). These results suggest that GRSF1 is directly binding and likely stabilizing *Dux* mRNA. Importantly, we did not observe any alteration in the level of pCHK1 upon *Grsf1* KD (*Figure 5g*; *Figure 5—source data 1*), excluding the possibility of *Dux* modulation through induction of RS in GRSF1-deficient cells. Of note, GSRF1 has been found to be phosphorylated in putative ATR and CHK1 sites following cellular stress in mass spectrometry based phospho-proteomic screens (*Mertins et al., 2014*), suggesting that direct phosphorylation by ATR/CHK1 might affect its function in *Dux* regulation. Although further studies will be required to explain the specific role of GRSF1 in the post-transcriptional regulation of *Dux* through ATR, these findings shed light on the link between RSR and *Dux* regulation. Furthermore, they support recent observations reporting the involvement of mRNA processing factors in the regulation of 2C-genes (*Rodriguez-Terrones et al., 2018*).

## ATR-activated ESCs gain expanded developmental potential

Our results so far indicated that ATR-activated ESCs exhibit similar transcriptional profile and characteristics of 2C-like cells. These findings led us to ask whether ATR-activated ESCs also gain expanded developmental potential similar to 2C-like cells, which are able to contribute to both embryonic and extra-embryonic tissues (*Choi et al., 2017*; *Ishiuchi et al., 2015*; *Macfarlan et al., 2012*).

To test this hypothesis, both *Atr*^Sec/Sec^ and *Atr*^+/+^ ESCs were cultivated in the absence or in the presence of APH and were subsequently differentiated in vitro toward trophoblast-like stem cells (TSCs) for three days, followed by terminal differentiation to TGCs upon withdrawal of fibroblast growth factor 4 (FGF4) and heparin for additional three days (*Figure 6a*; *Abad et al., 2013*). The RT-

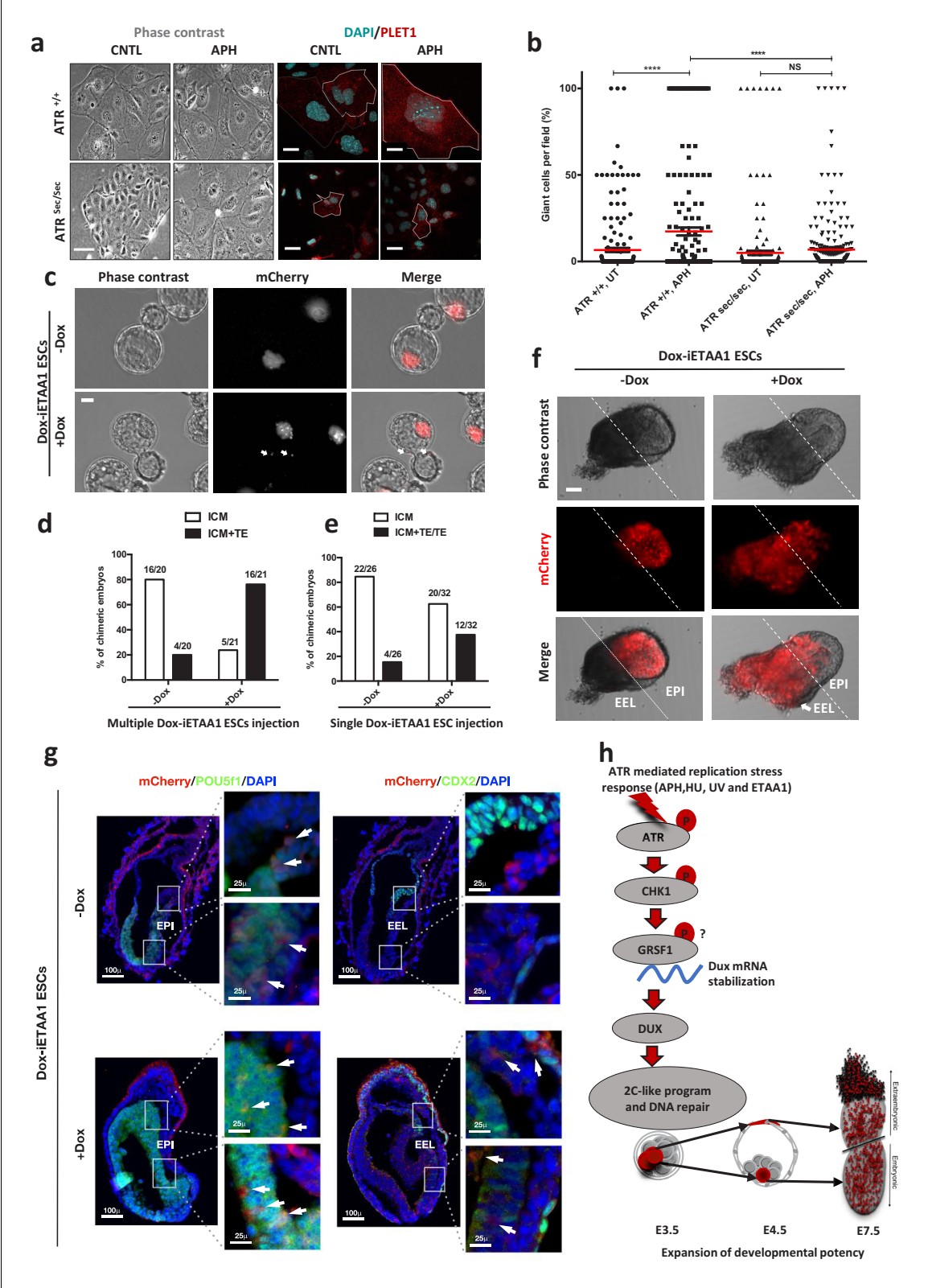

**Figure 6.** ATR-activated 2C-like cells gain expanded developmental potential in vitro and in vivo. (**a**) Phase contrast and immunofluorescence images of TGCs formed by in vitro differentiation of $Atr^{+/+}$ or $Atr^{sec/sec}$ ESCs treated with or without APH (bar = 25 μm). (**b**) Quantification of the number of TGCs detected in the conditions represented in a. (**c**) Images of blastocysts displaying the contribution of mCherry-labeled Dox-iETAA1 ESCs to the ICM and TE layers in the presence and absence of Dox (bar = 20 μm). (**d**) Bar plot showing the percentage of chimeric embryos in which injected mCherry-

*Figure 6 continued on next page*

*Figure 6 continued*

labeled Dox-iETAA1 ESCs could contribute to either ICM or ICM+TE with or without Dox. The ratios on top of each bar show the actual number of embryos that were analyzed. (**e**) Bar plot showing the percentage of chimeric embryos in which single-cell injection of mCherry-labeled Dox-iETAA1 ESCs could contribute to either ICM or ICM+TE/TE with or without Dox. The ratios on top of each bar show the actual number of embryos that were analyzed. (**f**) Images showing the contribution of injected mCherry-labeled Dox-iETAA1 ESCs to the epiblast (EPI) or extra-embryonic layers (EEL) of mouse embryos at E7.5 with or without Dox treatment (bar = 50 µm). (**g**) Immunostaining of mouse embryos at E7.5. Arrows in the left panel indicate the contribution of mCherry-labeled Dox-iETAA1 ESCs to the EPI (marked by POU5F1) in Dox-treated and untreated conditions. Arrows in the right panel indicate the contribution of injected mCherry-labeled Dox-iETAA1 ESCs to the EEL (marked by CDX2) only in Dox-treated condition. (lower magnification, bar = 100 µm, higher magnification, bar = 25 µm). (**h**) Schematic model defining a novel ATR-dependent transcriptional response to maintain the genomic integrity of developing embryos in response to RS. 1) ATR and CHK1-mediated RSR triggers the *Dux* mRNA accumulation through GRSF1 that in turn increases bipotent 2C-like cells by global transcriptional activation of 2C-specific genes including *Zscan4*. 2) ATR-induced bipotent ESCs extend their contribution to placental compartment. In the dot plot the mean is represented by a red line. Statistical significance compared to CNTL unless otherwise indicated. (*p≤0.05, **p≤0.01, ***p≤0.001, ****p≤0.0001, one-way ANOVA).

The online version of this article includes the following source data and figure supplement(s) for figure 6:

**Source data 1.** qPCR and TGCs quantification, and TE contribution counting.
**Figure supplement 1.** ATR-activated 2C-like cells gain expanded developmental potential.
**Figure supplement 2.** ATR-activated 2C-like cells gain expanded developmental potential.

qPCR results for the TGC specific marker, *Prl2c2*, revealed the highest expression in ATR-activated conditions, while $Atr^{Sec/Sec}$ cells could not significantly up-regulate this gene in the presence of APH (*Figure 6—figure supplement 1a*; *Figure 6—source data 1*). Moreover, *Prl2c2* basal level was significantly higher in WT cells compared to non-treated $Atr^{Sec/Sec}$ cells (*Figure 6—figure supplement 1a*; *Figure 6—source data 1*). These results were also consistent with the expression of Placenta-Expressed Transcript one protein (PLET1) and the number of TGCs generated in each condition (*Figure 6a and b*; *Figure 6—source data 1*). Next, to understand whether ATR-induced differentiation to TGCs is also mediated by DUX, we differentiated *Dux* KO and WT ESCs toward trophoblast cells upon ATR activation (*Figure 6—figure supplement 1b*). Significantly, ATR activation in *Dux* KO cells could not induce the formation of TGCs unlike WT cells, confirming the critical role of DUX in trophoblast-directed differentiation program upon ATR activation (*Figure 6—figure supplement 1c*; *Figure 6—source data 1*). These experiments indicate that the transition to 2C-like state is required for ATR-induced TGC differentiation.

Next, to define the cell fate potential of ATR-activated ESCs during embryonic development, we traced the fate of their progenies in chimeric blastocysts. To this end, ten mCherry fluorescent protein–labeled (mCherry-labeled) Dox-iETAA1 ESCs were treated with Dox and subsequently microinjected into each C57BL/6N recipient mouse morulae. Untreated Dox-iETAA1 ESCs were injected in parallel as control. The contribution of ESCs to the inner cell mass (ICM) and the trophectoderm (TE) layer of the blastocysts was then monitored 48 hr post-injection (*Figure 6c*). While both Dox-treated and untreated ESCs contributed to the ICM, strikingly, upon ATR activation, we found a statistically significant (Fisher's exact test p-value: 0.0005) increase in the number of chimeric embryos in which ESC progenies localized to both ICM and TE (*Figure 6d*; *Figure 6—figure supplement 1d*). Although the basal contribution level of untreated ESCs to the TE could be due to the previously reported effect of 2i (*Morgani et al., 2013*), overexpression of ETAA1-AAD significantly increased the number of embryos in which mCherry-labelled cells could contribute to the TE. Next, we asked whether activation of ATR through ETAA1-AAD expression is also able to impact on the number of cells contributing to the TE layer of the blastocyst. To this aim, we repeated this experiment and compared the number of cells in the TE layer of embryos injected upon Dox treatment to the one observed in CNTL. Significantly, ATR activation was able to increase about two-fold the average number of cells contributing to the TE layer (*Figure 6—figure supplement 1e*; *Figure 6—source data 1*). Finally, to understand whether ATR-dependent expanded developmental potential is mediated by DUX, we generated mCherry-labeled ETAA1-AAD Dox-inducible ESCs in a *Dux* KO background. As shown in *Figure 4—figure supplement 1k–m*, unlike WT cells that increased the expression of 2C-genes upon Dox treatment, *Dux* KO ESC were not able to activate these genes. Importantly, through the mouse chimera assay we found that activation of ATR was not able to significantly increase the number of mCherry-labeled Dox-iETAA1 *Dux* KO ESCs contributing to TE

layer upon Dox treatment, consistent with our in vitro results (*Figure 6—figure supplement 1f*; *Figure 6—source data 1*).

The expanded cell fate potential of RS-induced ESCs could originate from cells with bidirectional potential or a heterogeneous population containing cells that preferentially differentiate into embryonic or extraembryonic lineages (*Choi et al., 2017*). To distinguish between these two possibilities, we microinjected a single mCherry-labeled Dox-iETAA1 ESC upon Dox treatment into each C57BL/6N recipient morulae to generate chimeric blastocysts. Consistent with our previous findings, a considerable number of ATR-activated single cells were able to contribute to ICM+TE or TE, albeit with lower efficiency (Fisher's exact test p-value: 0.08) (*Figure 6e*; *Figure 6—source data 1*). To further validate these observations, we generated post-implantation chimeric embryos by microinjecting ten mCherry-labeled Dox-iETAA1 ESCs into each C57BL/6N morulae that were subsequently transferred to foster mothers. While WT ESCs contributed exclusively to the embryonic tissue (epiblast), ATR-activated cells contributed to both embryonic and extra-embryonic cell lineages in 50% (3 out of 6) of chimeric embryos at embryonic day (E) 7.5 (*Figure 6f and g*; *Figure 6—figure supplement 1g*; *Figure 6—figure supplement 2a and b*). These results indicate that ATR-activated ESCs gain expanded developmental potential, in contrast to ESCs, which are mostly restricted to generate embryonic cell types (*Figure 6h*).

These findings overall indicate that ATR-activated ESCs gain expanded developmental potential in vitro and in vivo, and that this effect is mediated by DUX, whose physiological role in the activation of cleavage-stage transcription program has been reported in previous works (*De Iaco et al., 2017*; *Hendrickson et al., 2017*; *Whiddon et al., 2017*).

## Discussion

The physiological relevance of the 2C-like transcriptional program has been limited to 2C-stage embryos, and thus the molecular regulation involved in reactivation of 2C-like cells in ESCs culture and its possible significance in vivo at later stages of embryonic development have remained elusive.

Here, we provide evidence that transition to bipotent 2C-like cells in ESCs culture is triggered by activation of ATR, a developmentally essential DDR gene, which plays a crucial role in the maintenance of stem cells both in embryonic and adult tissues (*Brown and Baltimore, 2000*; *Ruzankina et al., 2007*). First, we found a significant enrichment of $\gamma$H2AX$^+$ and *p*-CHK1$^+$ cells within Em$^+$ cell population. Second, inhibition of RSR by specific ATR or CHK1 inhibitors, but not p38 inhibitors, could markedly reduce the number of 2C-like cells in ESCs culture. Third, ATR-deficient Seckel and haploinsufficient CHK1 ESCs exposed to RS could not significantly induce 2C-specific genes. Fourth, activation of ATR through ETAA1-AAD overexpression could further increase the population of bipotent 2C-like cells in ESCs culture in the absence of RS. Finally, through a candidate-based screen approach, we unraveled the mechanistic basis of ATR-stimulated transition to bipotent 2C-like cells by showing that this response induces a 2C-like transcriptional program through post-transcriptional regulation of *Dux*, the key inducer of zygotic genome activation (ZGA) in placental mammals (*De Iaco et al., 2017*; *Hendrickson et al., 2017*; *Whiddon et al., 2017*).

Importantly, although DNA damage-induced cellular differentiation has been frequently shown in stem cells (*Santos et al., 2014*; *Schneider et al., 2013*), here we report for the first time to our knowledge that RS in ESCs leads to the transition to a more developmentally potent state that is required for the subsequent differentiation toward TGCs. As far as the molecular mechanism of ATR-dependent *Dux* level increase is concerned, our results show that ATR activity is required to promote *Dux* mRNA accumulation at the post-transcriptional level. This finding is consistent with previous observations showing that the mRNA processing machinery can regulate DUX4 mRNA level by promoting its degradation (*Feng et al., 2015*). Significantly, we discovered that GRSF1, a RNA binding protein involved in several aspects of mRNA metabolism (*Ufer, 2012*), directly and specifically binds to *Dux* mRNA and it is required for *Dux* mRNA accumulation in response to ATR activation. The links between GRSF1 and *Dux* are unexpected and will require further work to be fully dissected.

Elevated DUX level in turn increases bipotent 2C-like cells in ESC culture through global transcriptional activation of 2C-specific genes, including genome caretaker genes, *Zscan4* and *Tcstv1/3* (*Zhang et al., 2016*; *Zalzman et al., 2010*). In addition to their role in telomere elongation through activation of telomere sister chromatid exchange (*Nakai-Futatsugi and Niwa, 2016*; *Zalzman et al.,*

*2010*), *Zscan4* genes can also promote DNA repair by facilitating heterochromatin decondensation and DNA demethylation (*Akiyama et al., 2015*; *Eckersley-Maslin et al., 2016*; *Dan et al., 2017*). Therefore, ATR-mediated activation of *Zscan4* genes likely contributes to ESCs genomic integrity in response to RS. Intriguingly, the DUX4 genes were originally discovered in an attempt to identify the target genes of Helicase-like Transcription Factor (HLTF) (*Ding et al., 1998*), that was later found to act at replication forks upon RS (*Peng et al., 2018*). Unexpectedly, our findings also show that ATR activation in ESCs can trigger the generation of cells with bidirectional cell fate potential (*Figure 6h*). Activation of this pathway could act as a safeguard mechanism to ensure genome integrity of the developing embryo in response to RS by promoting DNA repair, on the one hand, or by diverting unrepaired ATR-activated ESCs to extra-embryonic tissues, on the other hand, thus limiting the incorporation of cells with unrepaired DNA in embryo proper tissues. These findings could in part explain the phenotype of *Atr*$^{sec/sec}$ mouse embryo, in which failure in proper activation of ATR in response to RS leads to the accumulation of γH2AX positive cells in embryo proper (*Murga et al., 2009*).

Overall our findings revealed that 1) ATR is a potent upstream driver of 2C-specific genes in ESCs culture; 2) 2C-like transcriptional program can also be activated at later stages of embryonic development (rather than being activated only at 2C-embryo stage) in response to exogenous RS; 3) 2C-like state induced by RS and ATR influences ESC fate and impacts on its plasticity and likely genome integrity. These findings suggest that the physiological role of 2C-like transcriptional program might not only be restricted to ZGA. Importantly, ATR$^{-/-}$ mice have been reported to develop up to the blastocyst stage (*Brown and Baltimore, 2000*; *de Klein et al., 2000*), suggesting that presence of ATR may not play an essential role in ZGA at 2C-stage embryo. This would be consistent with a recent finding showing that *Dppa2* and *Dppa4*, which are expressed until the onset of gastrulation, were found to be the direct regulators of DUX-driven zygotic transcriptional program (*Eckersley-Maslin, 2018*). This behavior would be compatible with the reactivation of 2C-like transcriptional program at later stages of development, concomitantly with a weakening of the suppressive mechanisms that keep 2C-genes repressed in response to endogenous or exogenous stimuli. However, although our findings indicate that ATR activation impacts on the cell fate even at later stages of embryonic development, we also clearly show that RS *per se* is not able to trigger main 2C-genes in more committed embryonic cells such as MEFs, suggesting an involvement of robust repressive mechanisms in suppressing this pathway in more differentiated cells.

It is tempting to speculate that reactivation of such ATR-dependent transcriptional program in more committed adult stem cells in which these suppressive mechanisms fail, could be responsible for the expression of genes that are involved in invasiveness, angiogenesis and immunosuppression shared by placenta and cancer cells (*Costanzo et al., 2018*; *Ferretti et al., 2007*). This would be consistent with the recently reported *Dux*/DUX4 reactivation in a diverse range of tumors (*Yasuda et al., 2016*; *Yoshimoto et al., 2017*; *Preussner et al., 2019*). In line with this, a recent report has shown that DUX4 level increases specifically in cancer cells, promoting their ability to downregulate major histocompatibility complex I antigen presenting molecules (*Chew et al., 2019*), a feature shared between trophoblast and cancer cells that allows escaping from immune surveillance mechanisms (*Costanzo et al., 2018*). Activation of extraembryonic features in response to ATR in early stages of cellular transformation could be a major outcome of RS induced by oncogene activity in cancer cells and would provide an additional rationale for the potent tumor suppressive effect of ATR inhibitors (*Costanzo et al., 2018*; *Bartek et al., 2012*; *Bartkova et al., 2005*; *Dobbelstein and Sørensen, 2015*; *Macheret and Halazonetis, 2015*; *Nieto-Soler et al., 2016*). Intriguingly, extraembryonic tissues have been recently shown to have similar epigenetic methylation profiles to cancer cells (*Smith et al., 2017*), highlighting an additional common feature between these two tissues.

In summary, our findings shed light on the endogenous and exogenous stimuli that could contribute to the cellular plasticity of ESCs and also provide a fundamental insight into the alternative mechanisms these cells exploit to respond to RS and maintain genome integrity.

# Materials and methods

## Animals

All mice used to generate ATR Seckel ESCs and MEF lines were bred and maintained under specific pathogen-free conditions. C57Bl/6J and 129P2/OlaHsd mice were purchased from Charles River Laboratories Harlan Italy (currently known as Envigo), respectively. Animals were kept in ventilated cages in standard 12 hr light-dark cycle.

## Cell culture

ESCs were grown in feeder-free culture condition and incubated at 37°C under 3% $O_2$ tension. ESC medium composed of KnockOut DMEM (ThermoFisher, 10829–018), 15% ESC qualified Fetal Bovine Serum (ThermoFisher, 16141–079), 2 mM L-glutamine, 1/500 home-made leukemia inhibitory factor (LIF), 0.1 mM non-essential amino acids, 0,1 mM 2-mercaptoethanol, 50 units/mL penicillin, 50 mg/mL streptomycin supplemented with the two inhibitors (2i); PD 0325901 (1 μM) and CHIR 99021 (3 μM) was used. MEFs were cultured in high glucose DMEM (Lonza, BE12-614F) supplemented with 10% North American FBS, 2 mM L-glutamine, 0.1 mM non-essential amino acids, 50 units/mL penicillin, 1 mM Sodium Pyruvate and 50 mg/mL streptomycin (MEF medium).

## ESCs derivation from ATR seckel and CHK1 haploinsufficient mice

To establish $Atr^{Sec/Sec}$ ESCs, $Atr^{+/Sec}$ heterozygous mice were crossed. Morulae were recovered and cultured overnight in KSOM medium (Millipore, MR-020P-5D) under mineral oil. Blastocysts were placed on MEFs feeder layer in ESC medium with 2i and LIF at 37°C under 5% $CO_2$. Upon ICM expansion, cells were passaged on feeder layer to obtain the first stock. The cells were then characterized by genotyping. To establish $Chk1^{+/-}$ ESCs lines, embryos at morula stage were cultured overnight in KSOM supplemented with 2i. On the following day, blastocysts were plated individually in 96-well plate and cultured in N2B27 medium supplemented with 2i and LIF. After 6–7 days, the ICM outgrowth was disaggregated using Accutase. Clumps of cells were expanded every second day, or when the size of colonies reached to the proper expansion level.

## MEFs generation

To produce MEFs, C57BL/6 and 129P2/OlaHsd mice were used. MEFs were prepared by mechanical disaggregation, trypsinization and seeding of embryos (E12-E13) in MEF medium after removal of the head, tail, limbs, and internal organs. Each trypsinized embryo was plated into a 10 cm dish. Once cells reached to 90% confluency (after 48 hr), they were frozen and stocked for the subsequent experiments.

## Treatments

All treatments, including APH, HU, UV, ATRi, ATMi, p38i and CHK1i, were added into fresh ESC medium and the cells were kept at 37°C under 3% $O_2$ tension overnight (16–17 hr) unless otherwise indicated. APH was used at concentrations ranging from 0.4 to 6 μM as indicated. Similarly, HU was used at 0.1 mM - 0.2 mM. ATMi (KU-55933) and ATRi (VE 822) were used at the concentrations of 10 μM and 1 μM respectively. The CHK1 inhibitors LY2603618 and UCN-01 were used at the concentrations of 200 nM and 100 nM, respectively. UV radiation exposure was performed at the dosage of 5 J/m$^2$ or 10 J/m$^2$. For induction of ETAA1-AAD expression, Dox was used at the concentration of 1 μg/mL for 48 hr. SB 203580 hydrochloride (Tocris Bioscience) and SB 239063 (Tocris Bioscience) were used at the concentration of 5 and 10 μM.

## RNA extraction, cDNA synthesis and qPCR

RNA was extracted using RNeasy Mini Kit (QIAGEN, 74104) and quantified by spectrophotometer (NanoDrop, ThermoFisher) after removal of DNA contaminants by DNAse I digestion (QIAGEN, 79254). cDNA was prepared from 2 μg of total RNA using SuperScript III reverse transcriptase kit (ThermoFisher, 18080–044) with Oligo(dT)20 Primer (ThermoFisher, 18418–020) and dNTP Mix (ThermoFisher, 18427–013) following manufacturer's instructions. qPCR assay was performed based on standard protocol using final working concentration of 1X SsoFast EvaGreen Supermix (Bio Rad, 1725201) or 1X LightCycler 480 SYBR Green I Master (Roche, 04707516001), 0.5 μM primer mix and

5 ng of cDNA. GAPDH, TBP or GUSB were used as internal controls to normalise the qPCR data following the $\Delta\Delta C_t$ method. For the list of oligos used in this study please refer to the *Supplementary file 8*.

## RNA isolation from embryos and cDNA preparation

Cryopreserved C57BL/6N morulae purchased from Janvier Labs (QUICKBLASTO), were thawed according to the manufacturer's protocol. After recovery, a group of healthy morulae were treated with APH at final concentration of 1.5 µM in KSOM culture medium for 4 hr while the rest were kept as control. The day after, the same number of morphologically healthy and synchronized blastocysts (22 and 7 blastocysts in the 1st and 2nd round, respectively) were selected from both groups. The APH concentration and the treatment interval had no effect on the viability of the embryos. RNA isolation was performed based on the previously published works (*Eckersley-Maslin, 2018*; *Costanzo et al., 2018*). Briefly, following the removal of zona pellucida in acidic tyrode's solution (Sigma-Aldrich, T1788), the pool of embryos was collected in 10 µl of lysis buffer containing 5X First Strand Buffer (from SuperScript III reverse transcriptase kit, ThermoFisher) supplemented with 0.1% Tween-20. Subsequently, embryos were mechanically ruptured by three cycles of freeze/thaw. As the quantity of isolated RNA is not detectable by NanoDrop spectrophotometer, the supernatant was directly used for cDNA preparation following centrifugation at 10621 g for 2 min. cDNA preparation and qPCR were performed following the aforementioned protocols.

## Flow cytometry (FACS)

ESCs were fixed and permeabilized using Cytoperm/Cytofix kit (BD Biosciences, 554714), and subsequently stained for 1 hr at room temperature with anti-H2AX-Phosphorylated (Ser139) antibody conjugated with Alexa Fluor 647 (Biolegend, 613407, RRID:AB_2114994), or Anti-p-CHK1 antibody conjugated with PE (Cell Signaling, 12268, RRID:AB_2797863) or Anti-ZSCAN4 (Merck Millipore, AB4340, RRID:AB_2827621) antibody conjugated with Anti-Rabbit ALexa488. Cells were washed and acquired on a FACSCalibur instrument (BD Biosciences, RRID: SCR_000401) or Attune NxT (Thermo Fisher Scientific). For Emerald, mCherry and GFP acquisition, cells were trypsinized, collected and subsequently acquired without fixation. Cell cycle profiles were acquired after incubation with DAPI (4',6-diamidino-2-phenylindole) overnight.

## Immunocytochemistry

Briefly, cells were fixed in 4% formaldehyde (Sigma-Aldrich) for 20 min and subsequently blocked for 1 hr in 10% FBS and 0.1% Triton-X100. Then, cells were incubated with primary antibody at 4°C overnight, followed by washes and incubation with secondary antibodies for 1 hr at room temperature. Next, samples were mounted and images were acquired with wide-field fluorescence microscope (Olympus AX70, Olympus). Acquired images were analyzed by ImageJ software (NIH, RRID:SCR_003070). For staining mouse embryos, blastocysts were fixed in 4% formaldehyde, followed by permeabilization for 10 min in 0.5% Triton X-100 and blocking for 1 hr at room temperature in 3% BSA and 0.1% Tween-20 in PBS. Then blastocysts were incubated with primary antibodies at 4°C overnight, before washes in 0.1% BSA in PBS and secondary antibodies incubation for 2 hr at room temperature, followed by a 15 min incubation with DAPI. Stained blastocysts were imaged with Leica TCS SP2 AOBS inverted confocal microscope.

Antibodies used in this study were Anti-Oct-3/4 (Santa Cruz, sc-5279, RRID:AB_628051), Anti-Nanog (Abcam, ab80892, RRID:AB_2150114), Anti-GFP (Abcam, ab5450, RRID:AB_304897), Anti-Plet-1 (Nordic MUbio, MUB1512P, RRID:AB_2827622), Anti-Cdx2 (Biocare Medical ACI 3144B, RRID:AB_2827623), Anti-mCherry (ChromoTek 5F8 a-RFP, RRID:AB_2336064.), Anti-ZSCAN4 (Merck Millipore, AB4340, RRID:AB_2827621), Anti-MERVL-GAG (Huabio, ER50102, RRID:AB_2636876), Anti-ETAA1 (Abcam, ab197017, RRID:AB_282762).

## Immunoblotting

Cells were trypsinized and washed with PBS and then lysed for 30 min on a rotating wheel at +4°C in RIPA buffer supplemented with protease/phosphatase inhibitor cocktail (Cell Signaling, 5872). Lysates were sonicated with a Bioruptor Sonication System (UCD200) at high power for 3 cycles of 30 s with one minute breaks. Lysates were centrifuged at 13000 rpm for 20–30 min and clear

supernatants were transferred to new tubes. Protein content was quantified using Bio-Rad protein assay according to manufacturer's instructions. For the detection of each protein, 35 µg of total protein extracts were loaded. Standard western blot was performed using following antibodies: Anti-ATR (Cell Signaling, 2790, RRID:AB_2227860), Anti-p-CHK1 (S317) (cell signaling, 12302, RRID:AB_2783865), Anti-ZSCAN4 (Merck Millipore, AB4340, RRID:AB_2827621), Anti-CHK1 (Santa Cruz, 8408, RRID:AB_627257), Anti-CHK2 (Millipore, 05–649, RRID:AB_2244941), Anti-γH2AX (Merck Millipore, 05–636, RRID:AB_309864), Anti-P53 (Cell Signaling, 2524S, RRID:AB_331743), Anti-p-P53 (S15) (Cell Signaling, 12571S, RRID:AB_2714036.), Anti-GAPDH (Abcam, ab9484, RRID:AB_307274), Anti Oct-3/4 antibody (Santa Cruz, sc-5279, RRID:AB_628051), Anti-VINCULIN (Sigma, V9131, RRID:AB_477629), Anti-ATM (Sigma, A1106, RRID:AB_796190), Anti-GRSF1 (Abcam, AB246330, RRID:AB_2827628.), Anti-PCBP2/hnRNP E2 (Abcam, AB236137, RRID:AB_2827629.), Anti-MERVL-GAG (Huabio, ER50102, RRID:AB_2636876), Anti-ETAA1 (Abcam, ab197017, RRID:AB_282762), Anti-SOX2 (Chemicon International, AB5603, RRID:AB_2286686).

## Drop-seq single cell mRNA sequencing

ESCs from CNTL, APH (6 µM) and APH-ATRi experimental conditions were resuspended in PBS-BSA and processed with a microfluidic device according to the DropSeq Laboratory Protocol V3.1 from McCarroll's Lab website [http://mccarrolllab.com/dropseq/]. For each condition, three distinct aliquots of collected emulsion, each containing 4000 beads underwent reverse transcription (RT) and fragment library preparation using Illumina Nextera XT Library Prep Kit. A distinct barcode was used for each library to allow subsequent demultiplexing of sequencing reads. mRNA sequencing was performed using an Illumina HiSeq2000 instrument, library fragments were sequenced at 50 base pairs (bp) in PE mode. Sequencing reads were aligned to UCSC Mouse Reference Genome version mm10 using STAR (version 2.5.3a), and processed according to DropSeq Alignment Cookbook V1.2 to generate a digital expression matrix of STAMPs (cells) for each experimental condition, which was then furtherly analyzed in the RStudio environment (R version 3.3.3, RRID:SCR_00432) using Seurat package V2.0 from Satija Lab [http://satijalab.org/seurat/] (RRID:SCR_007322). Seven-hundred highly expressing cells were selected in each condition; genes expressed in less than 3 cells and cells expressing less than 200 genes were pruned from the dataset. Data from CNTL and APH conditions (1399 cells) or CNTL, APH and APH-ATRi conditions (2096 cells) were merged to generate two datasets, which were log normalised and scaled. Expression mean and variance to mean ratio (LogVMR >0.5) were used to estimate data dispersion and select variable genes across the dataset. Respectively 2614 and 2768 genes (in the 2-conditions and 3-conditions datasets) were used in principal component analysis (PCA) to identify the appropriate number of components to include in data modeling. Top 15 components were selected to explain dataset complexity. A shared nearest neighbor (SNN) graph and smart local moving algorithm was used to perform cells clustering; t-SNE (t-stochastic neighbor embedding) analysis was used to reduce (PCA) dimensionality and generate plots of cells distribution.

## RNA sequencing library preparation and sequencing

RNA samples were quantified using Qubit 2.0 Fluorometer (Life Technologies, Carlsbad, CA, USA) and RNA integrity was checked with RNA Screen Tape on Agilent 2200 TapeStation (Agilent Technologies, Palo Alto, CA, USA). All RNA samples had a RIN score of 10. Ribo-Zero rRNA Removal Kit and TruSeq Stranded Total RNA library Prep kit was used to generate barcoded fragment libraries for RNA sequencing, following manufacturer's protocol (Illumina, Cat# RS-122–2101). Briefly, rRNA depleted RNAs were fragmented for 8 min at 94°C. First strand and second strand cDNA were subsequently synthesized. The second strand of cDNA was marked by incorporating dUTP during the synthesis. cDNA fragments were adenylated at 3'ends, and indexed adapters were ligated to cDNA fragments. Limited cycle PCR was used for library enrichment. The incorporated dUTP in second strand cDNA quenched the amplification of second strand, which helped to preserve the strand specificity. Sequencing libraries were validated using DNA Analysis Screen Tape on the Agilent 2200 TapeStation (Agilent Technologies, Palo Alto, CA, USA), and quantified by using Qubit 2.0 Fluorometer (Invitrogen, Carlsbad, CA) as well as by quantitative PCR (Applied Biosystems, Carlsbad, CA, USA).

Barcoded libraries were normalized, pooled and loaded on a single flowcell's lane of Illumina HiSeq4000 (RRID:SCR_016386) sequencer for cluster generation and sequencing, according to manufacturer's instructions. The samples were sequenced using a 2 × 150 Pair-End (PE) High Output configuration. Image analysis and base calling were conducted by the HiSeq Control Software (HCS) on the HiSeq instrument. Raw sequence data (.bcl files) generated from Illumina HiSeq was converted into FASTQ files and de-multiplexed using Illumina bcl2fastq program version 2.17. One mismatch was allowed for index sequence identification. On average ~29 million reads were generated for each sample and average insert size ranged between 204–231 for different libraries.

## RNA sequencing data processing

Transcript and gene level quantification was done with Kallisto (version 0.43.0) (*Bray et al., 2016*). The RefSeq (RRID: SCR_003496) NM and NR sequence collection as of 2016-11-21, was used to build the transcriptome index. Kallisto (RRID: SCR_016582) was run with the –bias option to perform sequence based bias correction on each sample. Additionally, all reads were mapped with the STAR spliced aligner (version 2.5.2b) (RRID: SCR_015899) (*Dobin et al., 2013*) to the GRCm38 (mm10) genome after trimming the reads to 100 bp with Trimmomatic (version 0.36) (RRID: SCR_011848) (*Bolger et al., 2014*). Repeat expression was quantified by counting fragments overlapping with the repeat annotation from *Choi et al. (2017)*, using FeatureCounts (RRID: SCR_012919) from the subread package (version 1.5.2) (*Liao et al., 2014*). FeatureCounts was run with the –p –B –C –primary options.

## RNA sequencing: gene level differential expression

Read counts for each sample estimated by Kallisto were imported into the R statistical environment (version 3.2.2) using tximport (version 1.1.2) (*Soneson et al., 2015*) (RRID: SCR_016752). The limma (version 3.24.15) (RRID: SCR_010943) package (*Ritchie et al., 2015*) was used to test for differential expression between CNTL and APH or CNTL and APH+ATRi treatments after TMM normalization and voom transformation (*Liao et al., 2014*). A linear model was fitted with the limma lmFit function and the moderated t-statistics was calculated with the eBayes function. Genes were defined as differentially expressed if they had FDR adjusted p-values<0.05 and |log$_2$ fold change| > 1.

The robust z-score for each gene in each sample was calculated using the median and mean absolute deviation (MAD) for each gene across samples, and standardizing expression with the following formula:

*Robust Z − score =* $(x - median)/MAD$ where x is the expression in TPM for a single gene in a given sample.

## RNA sequencing repeat analysis

Read counts for each sample based on the STAR (RRID:SCR_015899) alignment were imported into the R statistical environment (version 3.2.2) and summarized on the repeat subfamily level. edgeR (version 3.14.0) (RRID: SCR_012802) (*McCarthy et al., 2012*) was used to test for differential expression between CNTL and APH treatments, after TMM normalization. All repeat subfamilies were dropped where the count per million value did not reach one in at least two samples. A linear model was fitted with the edgeR glmFit function and a likelihood ratio p-value was calculated with the glmLRT function. Repeat subfamilies were defined as differentially expressed if they had FDR adjusted p-values<0.05 and |log2 fold change| > 1.

## RNA sequencing gene set enrichment analysis

Gene set enrichment was calculated using a mouse version of MSigDB (RRID:SCR_016863) available from http://bioinf.wehi.edu.au/software/MSigDB/ (Accessed September 12, 2016). The CAMERA method (*Wu and Smyth, 2012*) available in the limma package was used to check for significant enrichment of specific gene sets between the CNTL and APH treatments, using the same TMM normalized read counts as in the gene differential expression analysis. Additionally, the GSVA (*Hänzelmann et al., 2013*) package was used to transform gene level read counts per sample to a gene set level enrichment score per sample and for each MSigDB gene set. After this, the same limma analysis was carried out on the gene set enrichment values as for the gene level data. Gene

sets were defined as differentially expressed if they had FDR adjusted p-values<0.05 and |log$_2$ fold change| > 0.5.

Gene set enrichment for the Control and Dox treatment or Control and *Dux* KD+Dox treatment was calculated with the ROAST method available in the limma (version 3.24.15) package (*Hänzelmann et al., 2013*; *Wu et al., 2010*).

### RNA sequencing literature comparison

For all of the literature comparisons, the same transcript and gene level quantification methods and transcriptome annotations were used as for our data using the same limma package and methods to define differentially expressed genes between conditions. The Fisher-test was used to check for the significance of the overlaps in differentially expressed genes between our data and the literature datasets.

### Transcription factor binding site prediction

To predict transcription factor binding sites in the *Dux* promoter region, we extracted the promoter sequence around the Dux transcription start site (TSS), with 1000 nucleotides upstream and 100 nucleotides downstream of the TSS. We submitted this sequence to the JASPAR database (RRID: SCR_003030), and predicted putative binding sites using the JASPAR CORE set (*Khan et al., 2018*). We filtered out predicted binding sites with a score <5, and ranked the putative transcription factors regulating Dux based on the total number of binding sites predicted.

### Transfection

ESCs were plated in 2i plus LIF medium as described above with or without antibiotics according to the applied Lipofectamine (RNAiMAX or 2000, respectively). Lipofectamine (Sigma-Aldrich) was diluted in Opti-MEM (ThermoFisher) according to the manufacturer's protocol and incubated for 5 min at room temperature. siRNAs and MISSION esiRNAs (Sigma-Aldrich) were diluted in Opti-MEM to the opportune concentration and then added to the Lipofectamine emulsion at 1:1 ratio. Following incubation at room temperature (timing depends on the type of Lipofectamine), the mixture was added to the cells suspension in culture medium. The final concentration of *Dux* and *Trp53* siRNAs were 75 nM and 40 nM, respectively and the final concentration of MISSION esiRNAs were 60 nM. For each experiment a sample was transfected with MISSION siRNA Fluorescent Universal Negative Control, Cyanine 5 (for siRNA transfection) or esiRNA targeting RLUC (for MISSION esiRNA transfection) as negative controls at the same concentration used for the siRNA or esiRNA of interest. Gene expression was assessed 48 hr post transfection.

### Cloning, lentivirus production and induction

pLVX-EF1α-IRES-mCherry vector was co-transfected with plasmids encoding POL, REV, TAT, and the vesicular stomatitis virus envelope glycoprotein (VSV-G) into 80% confluent 293 T cells (RRID: CVCL_0063) using calcium phosphate precipitation in the presence of 25 mM chloroquine. The supernatant of transfected cells was collected every 24 hr for two days and concentrated using PEG-itTM virus precipitation solution and viral particles were resuspended in DMEM and frozen in small aliquots at −80˚C.

The Lenti-XTM Tet-On 3G Inducible Expression System was used to express ETAA1-AAD protein in ESCs. The cDNA of ETAA1-AAD (kind gift from Mailand lab, Novo Nordisk Foundation Center for Protein Research, Copenhagen, Denmark) was cloned in pLVX-TRE3G vector. Viral particles of pLVX-TRE3G-ETAA1-AAD and pLVX-EF1a-Tet3G vectors were separately produced (as mentioned above). ESCs were co-infected with both lentiviruses in the presence of 8 μg/mL polybrene, and subsequently infected cells were subjected to puromycin and neomycin selection for two weeks. To induce ETAA1-AAD expression, selected ESCs were treated with 1 μg/mL Dox for 48 hr. To generate *Dux* and luciferase overexpressing ESCs, pCW57.1-Luciferase (RRID:Addgene_99283) and pCW57.1-mDux-CA (RRID:Addgene_99284) plasmids were used for lentivirus production.

### siRNA-based screening

The esiRNA screening was performed on Tecan Freedom EVO 200 Automated workstation (Tecan Group Ltd., Switzerland, RRID:SCR_016771), equipped with RoMa (Robotic Manipulator Arm) and

LiHa (Liquid Handling Arm) with 1 ml dilutors. Briefly, ESCs were reversely transfected with esiRNAs and Lipofectamine RNAiMAX (Invitrogen 56532) in 96-well plates. Subsequently, cells were treated with APH or ATRi 36 hr post-transfection. Twelve hours after treatment, cell proliferation was assessed by CyQUANT Cell Proliferation Assay kit (ThermoFisher C35011), followed by cell lysis using Cells-to-CT Kit (ThermoFisher, AM1728). The Taqman qPCR assays were performed using Biomek FXP Automated Workstation (Beckman Coulter Life Sciences).

## In vitro differentiation and analysis

ESCs were differentiated toward TSC and TGCs as previously reported (*Abad et al., 2013*; *Tanaka et al., 1998*; *Ng et al., 2008*). Briefly, ATR Seckel or *Dux* KO ESCs along with WT ESCs were seeded on gelatin-coated 6-well plate in ESCs medium and 24 hr post treatment, medium was changed to trophoblast stem cell (TSC) differentiation medium, which contains: 30% RPMI 1640 (Lonza, BE12-167F) supplemented with 20% FBS, 1 mM pyruvate, 2 mM L-glutamine, 100 mM β-mercaptoethanol), 70% of conditioned medium from mitomycin-C-inactivated fibroblasts, 25 ng/mL FGF4 (R and D Systems, 235-F4-025) and 1 µg/mL heparin (Sigma-Aldrich, H3149). The medium was refreshed daily to maintain TSCs. To induce giant cell differentiation, TSCs were split at day two on gelatin-coated plate. After 24 hr, medium was changed to RPMI 1640 with 20% FBS, 1 mM pyruvate, 2 mM L-glutamine, 100 mM β-mercaptoethanol in the absence of heparin and FGF4. The medium was changed daily for 3 days.

## Mouse chimera assay

Mouse chimera assays were conducted at University of Copenhagen. Morulae were obtained from superovulated C57BL/6 prepubescent females. One (single cell experiments,) five (E4.5 embryos) or ten (E7.5 embryos) ESCs were injected into each morula after piercing their zona pellucida with beveled microinjection needle. Embryos were kept in M2 medium during the injection and afterwards in KSOM medium at 37°C under 5% $CO_2$. Morulae were either cultured for 48 hr to study ESC contribution at late blastocyst stage (E4.5), or transferred to pseudopregnant females and later dissected at E7.5. ESC contribution to the various embryonic layers were assessed by the localization of mCherry fluorescence signal.

For the APH-treated ESCs, frozen morula-stage embryos were purchased from Charles River and Janvier laboratories. Embryos were thawed 2 hr prior to injection and 4 to 8 ESCs were transferred into the perivitelline space by laser-assisted microinjection method. Afterwards, embryos were cultured for 24 hr at 37°C, 5% $CO_2$ until blastocyst stage (E3.5).

## RNA-pull down

RNA-pull down was performed using Pierce Magnetic RNA-Protein Pull-Down Kit (ThermoFisher, 20164) following provider's instruction using *Dux* 3'UTR synthetic RNA.

## Liquid chromatography–tandem MS (LC–MS/MS) analysis

The entire gel lane was processed with STAGE-diging protocol as described by Soffientini *et. al.* (*Soffientini and Bachi, 2016*). The entire protocol occurs in a p1000 tip (Sarstedt 70.762.100) filled at the orifice with a double C18 Empore Disk (3M, Minneapolis, MN) plug, named STAGE-diging tip. Briefly, after Coomassie staining, the entire lane was carefully cut into ~1 mm³ cubes and transferred into the STAGE-diging tip. These gel cubes were dehydrated with 100% acetonitrile (CAN, Carlo Erba 412392000) and rehydrated in 100 mM $NH_4HCO_3$ (Sigma A-6141) twice before being dehydrated by the addition of ACN. To ensure that the gel pieces do not create a sticky surface on the C18, all the solutions were added with a gel-loader tip. The removal of solutions was accomplished by centrifugation at 1800 rpm using the commercial tip box as holder. Reduction of protein disulfide bonds was carried out with 10 mM dithiothreitol (DTT, Nzytech MB03101) in 100 mM $NH_4HCO_3$ and subsequent alkylation was performed with 55 mM iodoacetamide (IAA Sigma I1149-25G), in complete darkness, in 100 mM $NH_4HCO_3$, at room temperature for 30 min. Both DTT and IAA were removed by centrifugation or by syringe as previously described. The gel pieces were rehydrated and dehydrated with 100 mM $NH_4HCO_3$ and ACN respectively prior to digestion. Gel pieces were rehydrated with 40 µL of Trypsin (12.5 ng/µL in 100 mM $NH_4HCO_3$, after few minutes 60 µL of $NH_4HCO_3$ were added and samples were incubated at 37°C o/n in a commercial tip box filled by

water on the bottom to ensure that buffer will not evaporate. The digestion solution was then forced through the double plug with a syringe and the flow through was collected. Samples were acidified with 100 µL of formic acid (FA, Fluka 94318) 0.1%, forced with the syringe and collected as flow-through to desalt the peptides. Peptides were eluted twice by adding 100 µL of a solution composed of 80% ACN, 0.1% FA, an extra step of extraction with 100% ACN was performed and then all the eluates were dried in a Speed-Vac and resuspended in 12 µL of solvent A (2% ACN, 0.1% formic acid). Four µL were injected for each technical replicate on the Q-Exactive –HF mass spectrometer.

## Mass spectrometry analysis

Mass spectrometry analysis was carried out by LC–MS–MS on a quadrupole Orbitrap Q Exactive HF mass spectrometer (Thermo Scientific). Peptide separation was achieved on a linear gradient from 95% Solvent A to 50% Solvent B (80% acetonitrile, 0.1% formic acid) over 20 min and from 50% to 100% Solvent B in 2 min at a constant flow rate of 0.25 µl minutes $^{-1}$ on a UHPLC Easy-nLC 1000 (Thermo Scientific), where the LC system was connected to a 25 cm fused-silica emitter of 75 µm inner diameter (New Objective), packed in house with ReproSil-Pur C18-AQ 1.9 µm beads (Maisch) using a high-pressure bomb loader (Proxeon). MS data were acquired using a data-dependent top15 method for HCD fragmentation. Survey full scan MS spectra (300–1750 Th) were acquired in the Orbitrap with 60,000 resolution, AGC target 1e6, IT 120 ms. For HCD spectra the resolution was set to 15,000, AGC target 1e5, IT 120 ms; normalized collision energy 28 and isolation width 3.0 $m/z$. Two technical replicates of each sample were carried out.

## Protein identification

For protein identification, the raw data were processed using Proteome Discoverer (version 1.4.0.288, Thermo Fischer Scientific). MS (*Ahuja et al., 2016*) spectra were searched with Mascot engine (RRID: SCR_014322) against uniprot_mouse_ database (80894 entries), with the following parameters: enzyme Trypsin, maximum missed cleavage 2, fixed modification carbamidomethylation (C), variable modification oxidation (M) and protein N-terminal acetylation, peptide tolerance 10 ppm, MS/MS tolerance 20 mmu. Peptide Spectral Matches (PSM) were filtered using percolator based on q-values at a 0.01 FDR (high confidence). Proteins were considered identified with two unique high confident peptides (*Käll et al., 2007*). Scaffold (version Scaffold_4.3.4, Proteome Software Inc, Portland, OR, RRID:SCR_014345) was used to validate MS/MS based peptide and protein identifications. Peptide identifications were accepted if they could be established at greater than 95.0% probability by the Peptide Prophet algorithm (*Keller et al., 2002*) with Scaffold delta-mass correction. Protein identifications were accepted if they could be established at greater than 99.0% probability and contained at least two identified peptides. Protein probabilities were assigned by the Protein Prophet algorithm (RRID:SCR_000286) (*Nesvizhskii et al., 2003*). Proteins that contained similar peptides and could not be differentiated based on MS/MS analysis alone were grouped to satisfy the principles of parsimony. Proteins sharing significant peptide evidence were grouped into clusters.

## β-Galactosidase staining

β-Galactosidase staining was performed following provider's instruction (Cell Signaling Technology, 9860).

## Ethics statement

The use and care of mice as described in this study was approved by the FIRC Institute of Molecular Oncology Institutional Animal Care and Use Committee in compliance with Italian law (D.lgs. 26/2014 and previously D.lgs. 116/92), which enforces Dir. 2010/63/EU (Directive 2010/63/EU of the European Parliament and of the Council of 22 September 2010, on the protection of animals used for scientific purposes) and by the Italian Ministry of Health experimental license 471/2015-PR. Mouse work at University of Copenhagen was conducted under the experimental license 2016-15-0201-00928 granted by the Danish Animal Inspectorate to A.J.L.C. All animal studies were performed in compliance with internationally accepted standards.

## Acknowledgements

We thank Anna De Antoni, Ambra Belpietro, and Eleonora Verga for technical support, Christopher Bruhn for helping with ATR Seckel mice colony maintenance and Federica Zanardi and Fabio Iannelli for helping with RNA design. The Niels Mailand lab (The Novo Nordisk Foundation Center for Protein Research, University of Copenhagen, Copenhagen, Denmark) for sharing ETAA1-AAD cDNA and advice. Alberto De Iaco and Didier Trono (School of Life Sciences, École Polytechnique Fédérale de Lausanne (EPFL), Lausanne, Switzerland) for sharing the *Dux* KO ESCs. Geppino Falco (Istituto di Ricerche Genetiche Gaetano Salvatore Biogem scarl, Ariano Irpino, Italy) for sharing *pZscan4*-Emerald ESCs. Austin Smith (Department of Biochemistry, University of Cambridge, Cambridge, UK) for sharing E14 ESCs. Yang Xu (University of California San Diego for sharing ATM KO ESCs. This work was funded by the Associazione Italiana per la Ricerca sul Cancro (AIRC), AIRC 5xmille METAMECH program, a European Research Council (ERC) consolidator grant (614541), the Giovanni Armenise-Harvard foundation award to VC, two research fellowships awarded by the Umberto Veronesi Foundation (FUV) and by FIRC - Fondazione Italiana per la Ricerca sul Cancro (The triennial 'Mario e Valeria Rindi' fellowship (18112)) to SA and three-year AIRC fellowship (23961) to NA awarded by FIRC. Work in AJLC lab was supported by grants from the Danish Cancer Society (KBVU-2014), Danish Council for Independent Research (Sapere Aude, DFF Starting Grant 2014), the European Research Council (ERC-2015-STG-679068) and the Danish National Research Foundation (DNRF115). The authors declare no financial conflict of interest related to this work.

## Additional information

### Funding

| Funder | Grant reference number | Author |
| --- | --- | --- |
| Fondazione Italiana per la Ricerca sul Cancro | FIRC 18112 | Sina Atashpaz |
| Fondazione Umberto Veronesi | | Sina Atashpaz |
| Associazione Italiana per la Ricerca sul Cancro | AIRC 5xmille METAMECH program | Vincenzo Costanzo |
| Giovanni Armenise-Harvard Foundation | | Vincenzo Costanzo |
| European Research Council | Consolidator grant 614541 | Vincenzo Costanzo |
| Associazione Italiana per la Ricerca sul Cancro | Fellowship 23961 | Negar Arghavanifard |
| Danish Cancer Society | KBVU-2014 | Andrés Joaquin López-Contreras |
| Danish Council for Independent Research | Sapere Aude, DFF Starting Grant 2014 | Andrés Joaquin López-Contreras |
| European Research Council | ERC-2015-STG-679068 | Andrés Joaquin López-Contreras |
| Danish National Research Foundation | DNRF115 | Andrés Joaquin López-Contreras |

The funders had no role in study design, data collection and interpretation, or the decision to submit the work for publication.

### Author contributions

Sina Atashpaz, Conceptualization, Formal analysis, Supervision, Funding acquisition, Validation, Investigation, Visualization, Methodology, Writing - original draft, Writing - review and editing, Performed most of the experiments including single-cell transcriptional profiling, high-throughput RNA sequencing, characterization of ATR Seckel and CHK1 haploinsufficient ESCs, cloning, candidate-based siRNA screening, generation of inducible ESCs, RNA pull-down and establishment of in vitro trophectoderm differentiation. Conceived, designed and analyzed the study and wrote the manuscript; Sara Samadi Shams, Conceptualization, Formal analysis, Supervision, Validation, Investigation, Visualization, Methodology, Writing - original draft, Writing - review and editing, Performed most of

the experiments including single-cell transcriptional profiling, high-throughput RNA sequencing, characterization of ATR Seckel and CHK1 haploinsufficient ESCs, cloning, candidate-based siRNA screening, generation of inducible ESCs, RNA pull-down and establishment of in vitro trophecto-derm differentiation. Conceived, designed and analyzed the study and wrote the manuscript; Javier Martin Gonzalez, Eliene Albers, Andrés Joaquin López-Contreras, Contributed to the generation of CHK1 haploinsufficient ESCs and the mice chimera experiments; Endre Sebestyén, Francesco Ferrari, Performed computational analysis on RNA-sequencing data; Negar Arghavanifard, Performed RNA isolation and qPCR on mice embryos, helped with the characterization of ATR Seckel and CHK1 haploinsufficient ESCs and establishment of in vitro differentiation protocol; Andrea Gnocchi, Helped with characterization of CHK1 haploinsufficient ESCs, performed in vitro differentiation experiments and analyzed the results; Simone Minardi, Performed and analyzed the Drop-seq single cell gene profiling; Giovanni Faga, Helped with siRNA screening; Paolo Soffientini, Angela Bachi, Performed mass spectrometry analysis; Elisa Allievi, Established ATR Seckel ESCs, generated MEFs and helped with mice embryo experiments; Valeria Cancila, Claudio Tripodo, Performed immunostaining of mouse embryos; Óscar Fernández-Capetillo, Shared ATR-Seckel mice and p53 KO cells and helped with technical analysis; Vincenzo Costanzo, Conceptualization, Supervision, Funding acquisition, Writing - original draft, Writing - review and editing, Conceived, designed and analyzed the study and wrote the manuscript

### Author ORCIDs
Sina Atashpaz https://orcid.org/0000-0003-0566-5629
Sara Samadi Shams https://orcid.org/0000-0002-1697-9343
Javier Martin Gonzalez http://orcid.org/0000-0002-7075-6028
Endre Sebestyén https://orcid.org/0000-0001-5470-2161
Negar Arghavanifard https://orcid.org/0000-0003-3039-7603
Andrea Gnocchi https://orcid.org/0000-0003-3290-9449
Simone Minardi http://orcid.org/0000-0001-7303-3821
Vincenzo Costanzo https://orcid.org/0000-0002-2920-9508

### Ethics
Animal experimentation: All mice used to generate ATR Seckel ESCs and MEF lines were bred and maintained under specific pathogen-free conditions. C57Bl/6J and 129P2/OlaHsd mice were purchased from Charles River Laboratories Harlan Italy (currently known as Envigo), respectively. Animals were kept in ventilated cages in standard 12 hours light-dark cycle. The use and care of mice as described in this study was approved by the FIRC Institute of Molecular Oncology Institutional Animal Care and Use Committee in compliance with Italian law (D.lgs. 26/2014 and previously D.lgs. 116/92), which enforces Dir. 2010/63/EU (Directive 2010/63/EU of the European Parliament and of the Council of 22 September 2010, on the protection of animals used for scientific purposes) and by the Italian Ministry of Health experimental license 471/2015-PR. Mouse work at University of Copenhagen was conducted under the experimental license 2016-15-0201-00928 granted by the Danish Animal Inspectorate to AJLC. All animal studies were performed in compliance with internationally accepted standards.

### Decision letter and Author response
Decision letter https://doi.org/10.7554/eLife.54756.sa1

---

## Additional files

### Supplementary files
• Supplementary file 1. The list of genes that are differentially expressed between each cluster cells and the rest of the population.

• Supplementary file 2. Single cells sequencing clusters GO analysis.

• Supplementary file 3. List of DEGs in APH induced ESCs.

- Supplementary file 4. Comparison of DEGs expressed in APH induced cells with published datasets.
- Supplementary file 5. Comparison of DE retroelements in APH induced cells.
- Supplementary file 6. List of Dux activators and supressors based on the screening experiment.
- Supplementary file 7. List of Dux RNA-bound factors identified through mass-spectrometry.
- Supplementary file 8. List of primers used in this study.
- Transparent reporting form

## Data availability

Raw sequencing reads for the bulk and single cell RNA-seq have been deposited in the NCBI BioProject database under accession number PRJNA415135 and PRJNA415187. All the proteomic data as raw files, total proteins and peptides identified with relative intensities and search parameters were loaded on Peptide Atlas repository (accession number http://www.peptideatlas.org/PASS/PASS01443) The source data underlying all main and extended figures are provided as a source data file.

The following datasets were generated:

| Author(s) | Year | Dataset title | Dataset URL | Database and Identifier |
| --- | --- | --- | --- | --- |
| Soffientini P | 2019 | Dux RNA-binding factors | http://www.peptideatlas.org/PASS/PASS01443 | Peptide Atlas, PASS01443 |
| Atashpaz S, Samadi S, Minardi S, Sebestyen E, Ferrari F, Costanzo V | 2019 | Mouse ES cell line transcriptome changes upon replication stress | https://www.ncbi.nlm.nih.gov/bioproject/?term=PRJNA415135 | NCBI BioProject, PRJNA415135 |
| Atashpaz S, Samadi S, Minardi S, Sebestyen E, Ferrari F, Costanzo V | 2019 | Mouse ES cell line single cell transcriptome changes upon replication stress | https://www.ncbi.nlm.nih.gov/bioproject/?term=PRJNA415187 | NCBI BioProject, PRJNA415187 |

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
