## [Decision Letter]

**Acceptance summary:**

This is a collaborative effort of different labs in which different cellular and genomic approaches have been used to gain insight into embryonic stem cell development, the role of replication stress and, importantly, the key role of ATR activation. The authors have started to identify downstream factors in the regulatory cascade, such as Dux4. Their future analysis will open new lines of research to unravel the intricate relationship between replication stress and stem cells development and differentiation mediated by ATR. Some of the observations will benefit from further study, such as the biological function of the 3,704 upregulated genes after APH treatment, which includes a good amount of retroviral elements, or the finding that 48% of APH-induced genes were transcriptionally repressed by ATRi in 2c-specific genes. This profile looks much like that of the chromatin assembly factor CAF-1 KD ESCs, and the authors argues that this may represent a way for cells to prevent replication stress, opening a connection with chromatin assembly during replication in stems cells. In conclusion, this manuscript presents results that are within the scope of the journal and of wide interest to the readership in multiple disciplines, not only in stem cell biology, embryogenesis and genome integrity, but also with respect to pathologies such as cancer. The data are well presented, controlled and the main conclusions are clearly justified.

[Editors’ note: the paper was accepted without revisions, so there is not an accompanying Author Response.]